# Federated Multimodal Fusion for Action Recognition Leveraging Vision-Language Embeddings and Spatio-Temporal CNNs

**Aditi Palit**                                                    *cs21d001@iittp.ac.in*
*Department of Computer Science*
*Indian Institute of Technology Tirupati*

**Kalidas Yeturu**                                                 *ykalidas@iittp.ac.in*
*Department of Computer Science*
*Indian Institute of Technology Tirupati*

**Reviewed on OpenReview:** *https://openreview.net/forum?id=AobzdtqiMe*

## Abstract

Federated learning (FL) for Video Action Recognition (VAR) faces significant challenges in balancing privacy preservation, communication efficiency, and model performance. This paper introduces *FLAMeST* (*F*ederated *L*earning for *A*ction Recognition with *M*ultimodal *e*mbeddings and *S*pacio-*T*emporal Fusion), a FL framework that synergizes Vision-Language Models (VLMs) and spatiotemporal CNNs to address these challenges. Unlike existing works that use BLIP (VLM) solely for caption generation, *FLAMeST* leverages BLIP in a dual manner. To enhance temporal modeling, complementary spatiotemporal features are extracted using a pre-trained 3D CNN (Slow network). These semantic (BLIP) and motion (Slow) embeddings are concatenated into a unified representation to train a lightweight Multi-Layer Perceptron (MLP). Within the FL paradigm, only the MLP parameters are shared with the server, ensuring raw video data and generated captions remain local. *FLAMeST* employs the FedAvg algorithm for model aggregation, achieving 99%($\downarrow$) lower communication overhead compared to full-model training. Experiments on UCF101 and HMDB51 datasets demonstrate the framework's robustness, achieving improved accuracies of 5.13%($\uparrow$) and 2.71%($\uparrow$), respectively, against the baseline.

## 1 Introduction

Video Action Recognition (VAR) plays a vital role in computer vision, with applications spanning human-computer interaction, surveillance, healthcare, and autonomous systems (Al-Faris et al., 2020; Javaid et al., 2024; Gumbs et al., 2022). Traditional VAR approaches leverage deep learning architectures—such as 3D Convolutional Neural Networks (3D-CNNs) (Ji et al., 2012; Tran et al., 2015), Recurrent Neural Networks (RNNs) (Ji et al., 2012; Tran et al., 2015; Yang et al., 2022),

and Transformers (Ulhaq et al., 2022), to model spatiotemporal dynamics from large-scale datasets like UCF101 (Soomro et al., 2012) and Kinetics (Carreira & Zisserman, 2017). However, these centralized training paradigms face limitations due to data privacy laws, e.g., GDPR (General Data Protection Regulation,(European Union, 2016)), data decentralization, and communication bottlenecks (AbdulRahman et al., 2020).

In real-world deployments, video streams from surveillance or wearable devices often contain sensitive user data, rendering inter-institutional sharing infeasible (Posner, 2008).

To address these challenges, Federated Learning (FL) has emerged as a promising solution by enabling decentralized training across clients without transferring raw data (Mammen, 2021; Kairouz et al., 2021). Clients update a shared global model through local training and only communicate parameter updates (McMahan et al., 2017). While this paradigm preserves privacy, most existing FL-based VAR methods rely on unimodal visual features and overlook the semantic richness of Vision-Language Models (VLMs) (Zhang et al., 2024). VLMs such as CLIP (Radford et al., 2021) and BLIP (Li et al., 2022) have shown superior performance in zero-shot and few-shot tasks by aligning visual and textual modalities through joint representations (Saha et al., 2024). These models offer rich semantic context—e.g., captions like "a person opening a door" can enhance action understanding when fused with visual features.

To address this issue of privacy-preserving VAR without compromising efficiency, we propose a framework for VAR in a cross-silos FL environment called *FLAMeST*, which stands for **F**ederated **L**earning for **A**ction Recognition with **M**ultimodal **e**mbeddings and **S**pacio-**T**emporal Fusion, that integrates spatiotemporal CNNs with VLM-based embeddings. In *FLAMeST* as shown in Figure 1, *each client uses a pre-trained BLIP model to generate a caption from a sampled video frame and then derives cross-modal embeddings by reprocessing the image-caption pair through BLIP's cross-attention module.* This dual use of BLIP, beyond standard caption generation, yields stronger semantic representations. In parallel, a SlowFast-3D CNN[1] (Feichtenhofer et al., 2019) extracts temporal features from keyframes and combines them with the VLM embeddings to form rich visual-level representations. The embeddings are concatenated and used to train a lightweight Multi-Layer Perceptron (MLP). To ensure communication efficiency, only the MLP weights are exchanged with the central server during aggregation, which drastically reduces communication overhead (Section 5.1, Table 1).

Our contributions are as follows:

- **Multimodal Embedding Fusion:** We introduce a novel hybrid embedding that combines BLIP-generated vision–language features with Slow-3D CNN outputs to capture both semantic and motion dynamics.

- **Communication Efficiency:** By training only the MLP in FL, our framework reduces communication overhead by 99.4% compared to full-model FL.

- *FLAMeST* achieves improvements of 5.13% and 2.71% over FL-based knowledge distillation (Jain et al., 2021) on the UCF101 and HMDB51 datasets, respectively.

Although VLMs are large-scale models, we assume clients can execute them in inference mode (Zhuang et al., 2023), enabling practical deployment while leveraging their strong semantic priors.

---

[1]For SlowFast, we used a Slow pathway with 8 frames as input.

Our primary objective is to explore the underutilized potential of VLMs in FL environments. While we instantiate FLAMeST for video action recognition on UCF101 and HMDB51, the underlying design is substantially more general. By freezing multimodal foundation backbones (BLIP and a 3D CNN) and federating only a lightweight prediction head, FLAMeST aligns with the broader trend in federated learning toward parameter-efficient tuning—where small heads, adapters, or prompts are collaboratively optimized while large backbones remain fixed across clients. This decoupled design is not tied to any specific dataset or architecture and can, in principle, extend to a range of vision or multimodal tasks that map pre-trained embeddings to downstream predictions.

## 2 Related Work

Human Action Recognition (HAR) tasks, particularly those relying on wearable sensors, generated considerable attention in both academia and industry (Gani et al., 2019; Wang et al., 2020; Hassan et al., 2018; Kalabakov et al., 2023), subsequently motivating the extension of such techniques to video-based applications.

Early VAR methods primarily relied on hand-crafted features such as motion-energy images (MEI) and motion-history images (MHI) to capture spatiotemporal dynamics (Bobick & Davis, 2001). Despite being computationally efficient, these methods exhibited limited robustness to variations in viewpoint and complex motion patterns (Bobick & Davis, 2001; Zhao et al., 2024).

**Centralized Action Recognition:** With deep learning, 2D CNNs like AlexNet and VGGNet (Krizhevsky et al., 2012; Simonyan & Zisserman, 2014) enabled robust spatial feature extraction from frames but lacked temporal modeling. To address this, two-stream networks emerged, combining RGB frames (spatial stream) and optical flow (temporal stream) (Alomar et al., 2024; Le et al., 2022). While these improved accuracy, they required separate optical flow computation, thus increasing complexity.

Subsequently, 3D CNNs such as C3D, I3D and SlowFast networks (Tran et al., 2015; Carreira & Zisserman, 2017; Feichtenhofer et al., 2019) enabled end-to-end spatiotemporal learning. Transformer-based models have also shown promise for VAR. Vision Transformers (ViTs) and Video Swin Transformers capture long-range temporal dependencies via attention mechanisms (Ulhaq et al., 2022; Liu et al., 2022), while hybrids like PSO-ConvNet Transformers combine CNN and transformer features for improved accuracy (Nguyen & Ribeiro, 2023). VideoMAE extends masked autoencoding to videos using spatiotemporal tube masking (Tong et al., 2022).

InternVideo2.5 (Wang et al., 2025) builds on the InternVideo2 architecture by introducing long-context modeling and hierarchical visual-token compression to efficiently process extended video sequences. It integrates a video encoder (for spatial–temporal features) with a language model backbone via a unified cross-modal adapter, enabling frame-level reasoning over long clips. The training follows a three-stage pipeline—progressive pretraining on image–text and short video–text pairs, followed by long-video alignment—to improve temporal coherence and cross-modal understanding.

However, these models are trained in centralized settings and focus on unimodal visual features, raising concerns over privacy, data sharing, and regulatory compliance (Kazakos et al., 2021; Akbari et al., 2021).

**Federated Learning for Action Recognition:** FL addresses privacy by enabling decentralized training. Several works have applied FL to HAR using wearable sensor data (Gani et al., 2019; Hassan et al., 2018). For video, most methods simplify the problem to image-level classification (Doshi & Yilmaz, 2022) or apply self-supervised learning to improve generalization (Rehman et al., 2022). Personalization and heterogeneity have been addressed through techniques like Meta-HAR (Li et al., 2021), FedCLAR (Presotto et al., 2022), and FedMAT (Shen et al., 2022), which balance shared and user-specific learning. An activity recognition framework in FL is proposed in (Yang et al., 2024), where both global (modality-agnostic) and private (modality-specific) classifiers are learned collaboratively across clients. This design effectively separates shared and unique modality characteristics through adversarial training. Architecture-aware FL approaches like FedConv (Xu et al., 2023) study CNN configurations under heterogeneity have also been explored. Addressing the limited computational capacity of edge devices, (Jain et al., 2021) proposed a knowledge distillation (KD) strategy involving two teacher models to facilitate efficient model deployment in an FL environment. This hierarchical distillation pipeline introduces a teaching assistant model as an intermediary, effectively bridging the gap between the complex teacher and the constrained student. For few-shot learning under FL, (Tu et al., 2024a) employed CNNs to capture spatiotemporal cues and applied meta-learning for improved generalization. FedVision enabled FL for object detection with YOLOv3 (Liu et al., 2020), while recent transformer-based FL work targets video anomaly detection (Doshi & Yilmaz, 2023).

Although these methods support video or multimodal learning in FL, they do not incorporate vision-language models (VLMs), missing the benefits of semantic context.

**Vision-Language Models and FL Integration:** CLIP and BLIP (Radford et al., 2021; Li et al., 2022) demonstrate strong image-text alignment for zero-shot tasks, but their application in video tasks remains limited. A few recent efforts addressed this gap—for example, (Zhuang et al., 2023) analyzes how foundation models can be incorporated into FL, outlining both opportunities for collaboration and the associated technical hurdles. ActionCLIP (Wang et al., 2023), which adapts VLMs like CLIP for video VAR by leveraging the semantic alignment between visual and textual modalities. Unlike traditional methods that treat action labels as discrete classes, ActionCLIP treats them as rich textual descriptions. JoVALE (Son et al., 2024) combines BLIP with person detection and audio-visual fusion, but the added modalities increase computational overhead.

To address these gaps, we propose *FLAMeST*, a framework that combines BLIP-based vision-language embeddings with spatiotemporal CNN features for efficient, privacy-preserving FL in VAR.

Our approach achieves competitive accuracy on UCF101 and HMDB51 while significantly reducing communication overhead.

## 3 FLAMeST Framework

The pipeline for the *FLAMeST* framework is shown in Figure 1. We consider an FL set-up comprising $N$ clients $\{C^1, \ldots, C^N\}$, where each client $C^i$ holds a private labeled video dataset. $\mathbf{V}^i_j$ represents the $j$th video clip of the $i$th device.

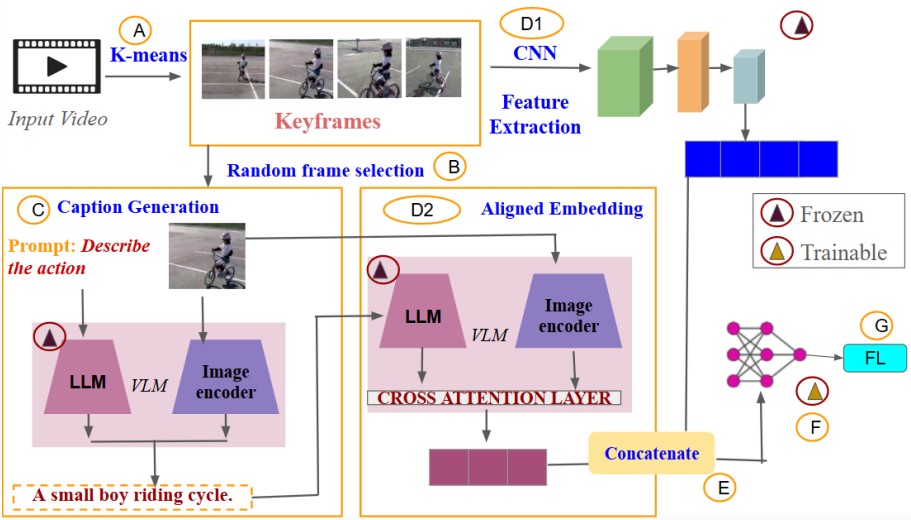

Figure 1: Overview of the *FLAMeST* pipeline. (A) Keyframes are selected using k-means clustering. (B-C) A keyframe is sampled and captioned using a VLM. (D1) All keyframes are passed through a 3D CNN; (D2) the sampled keyframe-caption pair is reprocessed for cross-modal embeddings. (E-F) CNN and VLM features are concatenated to train a lightweight MLP, which participates in the FL. Only the MLP parameters are updated and sent to the server, which aggregates via FedAvg and redistributes the global MLP (step G). The foundation models remain frozen and are not transmitted, reducing communication overhead (Supplementary Figure A-1).

$$\mathcal{D}^i = \{(\mathbf{V}_j^i, y_j^i)\}_{j=1}^{M^i}$$
$$\text{where } |D^i| = M^i. \tag{1}$$

Each video sample consists of $n_j^i$ RGB frames of spatial dimension $H \times W$ where the associated action class $y_j^i$ belongs in a label space of $K$ action classes.

$$\mathbf{V}_j^i \in \mathbb{R}^{n_j^i \times H \times W \times 3}$$
$$y_j^i \in \{1, \dots, K\} \tag{2}$$

### 3.1 Key Frame Selection via Clustering

To reduce the temporal redundancy in video sequences, we apply *k-Means* based clustering in a learned feature space to identify a reduced set of informative keyframes. Across all devices and their data elements, we set $m_j^i = k$, where $k$ is the number of clusters. We experimented with $k \in \{2, 3, 5, 8\}$ and selected k=3 using the elbow method and silhouette confidence intervals. This makes the keyframe selection systematic and reproducible, rather than heuristic. Method details

are presented in Supplementary Section A.1, Algorithm A-1.

$$\mathbf{F}_{\text{key},j}^i = \{\mathbf{f}_1^i, \ldots, \mathbf{f}_{m_j^i}^i\}$$
$$\text{where} \quad \mathbf{f}_l^i \in \mathbb{R}^{H \times W \times 3}, l \in \{1, \ldots, m_j^i\} \tag{3}$$

### 3.2 Random Key Frame Sampling

From the set of keyframes $\mathbf{F}_{\text{key},j}^i$ a single representative frame $\tilde{\mathbf{f}}_j^i$ is sampled uniformly at random.

$$\tilde{\mathbf{f}}_j^i \sim \mathcal{U}(\mathbf{F}_{\text{key},j}^i)$$
$$\text{where} \quad P(\tilde{\mathbf{f}}_j^i) = \frac{1}{m_j^i} \tag{4}$$

### 3.3 Multimodal Embedding Generation via VLM

The sampled frame $\tilde{\mathbf{f}}_j^i$ is first passed through a VLM $\Phi_{\text{VLM}}$ to generate a descriptive caption $\mathbf{t}_j^i$.

$$\mathbf{t}_j^i = \Phi_{\text{VLM}}(\tilde{\mathbf{f}}_j^i), \tag{5}$$

Subsequently, the caption $\mathbf{t}_j^i$ and the frame $\tilde{\mathbf{f}}_j^i$ are fed into the VLM cross-attention module to produce a joint vision-language embedding, i.e, the [CLS] token embedding of the last hidden layer is extracted (Algorithm 1). Without loss of generality, we consider the BLIP VLM.

$$\mathbf{e}_{\text{VLM},j}^i = \Phi_{\text{VLM}}^{\text{cross}}(\tilde{\mathbf{f}}_j^i, \mathbf{t}_j^i) \in \mathbb{R}^{d_e},$$
$$\text{where} \quad d_e = 1 \times 768. \tag{6}$$

### 3.4 Visual Embedding via CNN Backbone

In parallel, the set of keyframes $\mathbf{F}_{\text{key},j}^i$ is processed using a 3D CNN-based backbone $\Phi_{\text{CNN}}$ (e.g., ResNet-3D, I3D, Slow) to obtain a visual-only embedding. Without loss of generality, we consider the *Slow* -3D CNN model.

$$\mathbf{e}_{\text{CNN},j}^i = \Phi_{\text{CNN}}(\mathbf{F}_{\text{key},j}^i) \in \mathbb{R}^{d_v},$$
$$\text{where} \quad d_v = 1 \times 2048. \tag{7}$$

The embeddings are obtained from the last dense layer of the CNN model (Algorithm 1).

### 3.5 Joint Feature Representation

The final multimodal feature vector is formed by concatenating the VLM-based and CNN-based embeddings. We also experimented with a gated-attention-based fusion technique. However, the performance is not any better than simple concatenation (The results are provided in Section 6.5).

$$\mathbf{e}_j^i = [\mathbf{e}_{\text{VLM},j}^i; \mathbf{e}_{\text{CNN},j}^i] \in \mathbb{R}^{d_e + d_v}$$
$$\text{where dimensionality of } e_j^i \text{ is } 1 \times 2816. \tag{8}$$

Each client prepares its transformed data set,

$$\mathcal{D}^i_{\text{VLM,CNN}} = \{(\mathbf{e}^i_j, y^i_j)\}^{j=M^i}_{j=1} \tag{9}$$

The CNN ($\mathbf{e}^i_{\text{CNN},j}$) and VLM ($\mathbf{e}^i_{\text{VLM},j}$) embeddings often differ in dimensionality, making element-wise operations like the dot product infeasible. While projecting them to a common space is possible, it may introduce complexity and risk of information loss. To retain the full representation of both modalities without additional transformations, we adopt simple concatenation for fusion.

---

**Algorithm 1** High-level steps for obtaining VLM and CNN embeddings on each client

---

1: Determine keyframes (EQ. 3)
2: Determine a representative frame (EQ. 4)
3: Obtain caption for the frame (EQ. 5)
4: Obtain text-visual cross-embedding for the caption text and the corresponding frame (EQ. 6)
5: Obtain CNN embedding for the representative frame (EQ. 7)
6: Obtain combined embedding (EQ. 8)
7: Prepare a transformed data set (EQ. 9)

---

### 3.6 Local Training and Federated Aggregation at Server

On each client $C^i$ train a local multi-layer perceptron (MLP) classifier $g^i(\cdot)$ parameterized on $\mathbf{w}^i$ [2] on the transformed dataset $\mathcal{D}^i_{\text{VLM,CNN}}$.

$$g^i(\mathbf{w}^i) : \mathbb{R}^{d_e+d_v} \to \mathbb{R}^K \tag{10}$$

Each client trains $g^i(\cdot)$ by minimizing the cross-entropy loss using Stochastic Gradient Descent (SGD) as,

$$\mathcal{J}(\mathbf{w^i}) = \frac{1}{|\mathcal{D}|} \sum_{(e,y)\in\mathcal{D}} \mathcal{L}_{\text{CE}}(g(e), y)$$
$$\mathbf{w}^i_{t+1} = \mathbf{w^i}_t - \eta\nabla_{\mathbf{w^i}}\mathcal{J}(\mathbf{w^i}_t) \tag{11}$$

where $D$ is $D^i_{\text{VLM,CNN}}$, $g$ is $g^i(\mathbf{w}^i)$, $\mathcal{L}_{\text{CE}}(\cdot,\cdot)$ denotes the cross-entropy loss and $\eta$ is the learning rate. After local training, each client transmits its updated parameters $\mathbf{w}^i$ (EQ. 11) to a central server. The server aggregates these received models by weighted averaging of the models' parameters (EQ. 12, Algorithm 2).

$$\mathbf{w}^*_{(t+1)} = \sum^N_{i=1} \alpha^i \mathbf{w}^i_{(t)}, \quad \alpha^i = \frac{|\mathcal{D}i|}{\sum^N_{j=1}|\mathcal{D}^j|} \tag{12}$$

Where $\alpha$ corresponds to the data proportion of each client. It denotes the weight given to each client during model aggregation. The aggregated global model $\mathbf{w}^*_{(t+1)}$ is broadcast to all clients, and the training proceeds for $T$ communication rounds.

---

[2] $\mathbf{w}^i$ is $\mathbf{w}^i_{(t)}$ at time t

Each client has a pre-trained VLM and a CNN model, both frozen during training and inference. **The only trainable component is the MLP**, which participates in the FL cycle.

An alternative is to employ recent video-language captioners such as Qwen-VL or Cosmos-Reasoning. However, deploying multi-billion parameter captioners on each FL client is infeasible due to memory and latency constraints, and server-side captioning would require uploading raw video, violating privacy. Our *FLAMeST* pipeline is lightweight, client-friendly, and privacy-preserving. Exploring distilled captioners in FL is an avenue for future work.

---

**Algorithm 2** Federated Learning with federated averaging

---

1: Initialize server model to random weights $w_0^*$
2: **for** $t \in [1 \dots T]$ **do**
3:    **for** $i \in [1, \dots, N]$ **do**
4:       Initialize client weights (EQ. 10)

$$w^i = w_{(t-1)}^*$$

5:       Prepare (or update) client data set (Algorithm 1)
6:       Minimize client specific loss (EQ. 11)

$$w^i = \underset{w}{\mathrm{argmin}} \ \mathcal{J}^i(w)$$

7:    **end for**
8:    Perform federated averaging on server (EQ. 12) to obtain $w_{(t)}^*$
9: **end for**

---

## 4 Experimental Set-Up

We design a comprehensive experimental framework to evaluate the performance of *FLMeST* for VAR. Table 1 shows the details regarding each model and embedding size. All the experiments are done on NVIDIA A100. All references to BLIP embeddings in this work pertain to the cross-attention embeddings derived from the visual-semantic alignment within the BLIP model.

**Benchmark Datasets and Data Partitioning Strategy:** The experiments in this study are conducted on two widely recognized benchmark datasets: UCF101 (Soomro et al., 2012) and HMDB51 (Kuehne et al., 2011). UCF101 comprises 13,320 video clips spanning 101 action categories. The dataset is split into 10,619 training and 2,701 testing samples. HMDB51 consists of 6,766 video clips categorized into 51 action classes. The dataset is partitioned with 5,413 videos for training and 1,353 videos for testing, following an 80:20 train-test split. To simulate real-world Non-IID conditions, we partition data across clients using a Dirichlet distribution $\mathrm{Dir}(\beta)$, where a smaller $\beta$ indicates higher label skew. For baseline comparisons, we use IID splits ($\beta = 1$), while Non-IID settings use $\beta = 0.6$ to evaluate robustness under data heterogeneity.

**Benchmarking Methods and Feature Extractors:** We use two popular vision-language models (VLMs): BLIP-2 (Li et al., 2022) and CLIP-ViT/L-14 (Radford et al., 2021). For benchmarking, we evaluate four spatiotemporal models: (1) ResNet-3D (Tran et al., 2015) as a 3D CNN baseline; (2) I3D (Carreira & Zisserman, 2017), which inflates 2D kernels into 3D; (3) SlowFast (Feichtenhofer

et al., 2019) using an 8-frame Slow pathway to capture appearance and motion; (4) VideoMAE (Tong et al., 2022), a masked autoencoder-based ViT for video representation, and InternVideo2.5 (Wang et al., 2025), a VLM for video QA. All models use public code and pretrained weights from PyTorchVideo (Fan et al., 2021), Torchvision, and HuggingFace. Feature vectors are extracted from the final convolutional layer before classification to capture spatiotemporal semantics. In addition to the feature extractors discussed above, we benchmark our method against the work of (Jain et al., 2021)[3] We have also considered the recent method ActionCLIP by (Wang et al., 2023) in a non-FL setting that uses a CLIP model for action recognition on unseen classes. We do not compare our method with existing FL-based VAR approaches such as FL for Driving Action Recognition (DAR) (Doshi & Yilmaz, 2022), which targets driver-specific actions using 2D CNNs and FedGKT (He et al., 2020) for communication efficiency. Their task-specific focus differs from our objective of general-purpose action recognition. Similarly, FSAR (Guo et al., 2023) is another VAR method in the FL setting, but it operates on skeleton-based datasets rather than video data, making direct comparison with our approach inappropriate. Few-shot FL methods (Tu et al., 2024b) are also excluded, as they address a different problem setting. However, we consider the extension of our framework to few-shot FL-based VAR as an interesting direction for future work.

**Client-Side Training Protocol:** In the proposed FL framework, foundation models remain frozen during training and inference and do not participate in FL communication. Each client trains a local MLP classifier, whose parameters are shared with the server during aggregation. By default, the MLP comprises two hidden layers (512 and 256 neurons) and an output layer matching the number of action classes. The input layer aligns with the dimensionality of the frozen backbone embeddings. Clients train locally for five epochs per round using a batch size of 128. Optimization is performed using Adam with a learning rate of 0.01 and weight decay of $1 \times 10^{-4}$, with cross-entropy loss as the loss objective. Additional MLP configurations are explored in Section 6. Unless stated otherwise, experiments use four clients in a cross-silo IID-FL setting over 80 rounds.

**Model Aggregation, Client Update and Evaluation Metrics:** The model aggregation is accomplished via FedAvg (McMahan et al., 2017), which simply averages model weights across participating clients. For client-side model update, we have considered two methods, FedProx (Li et al., 2020) and FedDyn (Acar et al., 2021), apart from simple gradient update, which were compared in ablation studies (Section 6).

Performance is assessed using two primary metrics. The top-1 classification accuracy of the *global model* and communication efficiency are measured as the number of model parameters exchanged per communication round.

**Resource Usage and Storage Requirements:** We use the VLM BLIP[4] model for generating the caption and subsequently visual-semantic cross embeddings. For the 3D-CNN backbone, we used the Slow-3D CNN model[5], which is part of Facebook AI's PySlowFast framework. An analysis of resource usage during inference mode shows that the Slow-3D CNN model recorded a maximum memory allocation of 319.51 MB per point, while the BLIP model utilized 1023.97 MB per point as inspected using *watch*[6] and *torch*.[7] The execution times for Slow-3D CNN and BLIP are 1.315

---

[3]The code for this paper was not publicly available. Therefore, the accuracy values reported in our work are those quoted directly from the original paper.

[4]Salesforce/blip-image-captioning-base

[5]https://pytorch.org/hub/facebookresearch_pytorchvideo_resnet

[6]watch -n 1 nvidia-smi

[7]torch.cuda.max_memory_allocated()/(1024**2)

- **Parameter reduction by using only the MLP in FL :**

$$= \frac{\overbrace{(\text{BLIP} + \text{Slow}}^{279,868,172} + \text{MLP}) - \text{MLP}}{(\text{BLIP} + \text{Slow} + \text{MLP})}$$

$$= \frac{(279,868,172 + 1,573,889) - 1,573,889}{279,868,172 + 1,573,889}$$

$$= 0.994 \quad \text{or} \quad 99.4\% \qquad (13)$$

- **Parameter reduction compared to Knowledge Distillation (KD) :**

$$= \frac{\text{KD} - \text{Our}}{\text{KD}}, \quad \text{Our} = \text{MLP}$$

$$= \frac{11,689,512 - 1,573,889}{11,689,512} \qquad (14)$$

$$= 0.865 \quad \text{or} \quad 86.5\%$$

seconds and 1.362 seconds, respectively, when processing a single input point. The storage requirement[8] for BLIP is approximately 941.44 MB, the Slow-3D CNN model is 131.85 MB, and the MLP model is 6.02 MB (Refer to Table 1).

## 5 Results and Evaluation

In this section, we conduct a quantitative evaluation of *FLAMeST*, focusing on its communication overhead and performance in both FL and centralized learning setups. We compare *FLAMeST* against various foundation model extractors to assess its effectiveness and efficiency in these different learning environments.

### 5.1 Communication Overhead

We assume that each participating client possesses sufficient storage and computational resources to host pre-trained foundation models, such as BLIP, CLIP, and 3D-CNN. Table 1 outlines the parameter count of the models employed. We propose a lightweight communication strategy to overcome the significant communication overhead typically associated with federated training of large-scale models. Instead of exchanging the full parameter sets of these massive foundation models, we freeze the pre-trained model's backbone during local training and extract fixed embeddings from it. A lightweight MLP classifier is trained on these embeddings. This is the only component that solely participates in the FL optimization loop. This decouples the representation learning phase from the federated aggregation step, substantially reducing communication costs.

As shown in Table 1, the number of parameters involved in FL is smaller than the size of the underlying foundation models. For instance, while the BLIP+Slow model comprises approximately 279 million parameters, only 1,573,889 parameters from the MLP are involved in FL communication, resulting in a parameter transfer reduction of approximately 99.4% (EQ. 13). Similarly, compared to the baseline KD, the parameter transfer reduction is approximately 86.5% (EQ. 14). This selective participation significantly enhances the communication efficiency of the proposed framework without compromising representation quality. Freezing the foundation models strikes a balance between performance and efficiency, preserving the strong semantic priors of these models while avoiding the prohibitive cost of fine-tuning and communicating billions of parameters across clients.

---

[8]The Floating Point (FP) is 32

Table 1: Model Comparison: Parameters and Input Dimensions for Feature Extractors and FL Models, where the last column refers to the learnable parameters participating in FL. Details are presented in Section 4.

| S.No | Feature Extractor Models | Input Frame | Base Model Parameters | Input Dimension | Parameters in FL |
|---|---|---|---|---|---|
| 1 | BLIP (Li et al., 2022) | 1 | 247,414,076 | 768 | 525,311 |
| 2 | CLIP (Radford et al., 2021) | 1 | 151,277,313 | 1024 | 656,3855 |
| 3 | KD **(Baseline)** (Jain et al., 2021) | 8 | 49,482,360 | - | **11,689,512** |
| 4 | ResNet3D (Tran et al., 2015) | 3 | 33,371,472 | 512 | 394,241 |
| 5 | I3D (Carreira & Zisserman, 2017) | 8 | 28,043,472 | 2048 | 1,180,673 |
| 6 | Slow (Feichtenhofer et al., 2019) | 8 | 34,566,488 | 2048 | 1,180,673 |
| 7 | VideoMAE (Tong et al., 2022) | 16 | 86304869 | 786 | 525,313 |
| 8 | BLIP (Text Only) | 1 | 247,414,076 | 768 | 525,313 |
| 9 | BLIP + I3D | 9 (1+8) | 275,457,548 | 2816 | 1,573,889 |
| 10 | BLIP + ResNet3D | 4 (1+3) | 280,785,548 | 1280 | 797,457 |
| 11 | BLIP + Slow (*FLAMeST*) | 9 (1+8) | **279,868,172** | 2816 | **1,573,889** |
| 12 | BLIP (Text Only) + ResNet3D | 4 (1+3) | 280,785,548 | 1280 | 797,457 |
| 13 | BLIP (Text Only) + I3D | 9 (1+8) | 275,457,548 | 2816 | 1,573,889 |
| 14 | BLIP (Text Only) + Slow | 9 (1+8) | 279,868,172 | 2816 | 1,573,889 |
| 15 | ActionCLIP (non-FL) (Wang et al., 2023) | 3 | 150,000,000 | - | - |
| 16 | InternVideo2.5 (Wang et al., 2025) | 8 | 8,075,422,270 | 1024 | 682,085 |

## 5.2 Comparison of *FLAMeST* and other Feature Extractors

Table 2-(A, B) presents a comparative study of different VLMs, various CNN and transformer-based feature extractors in FL training settings on the UCF101 and HMDB51 datasets. For UCF101, the highest accuracy is achieved by the combination of embeddings extracted from the Slow model along with the cross-attention embeddings obtained from the BLIP model. Our method achieves an improvement of 5.13% against the baseline (Table 2-A, row 14, row 3).

The Slow model, being a 3D CNN, effectively captures the spatial-temporal aspects of video sequences. In contrast, the BLIP model integrates visual and textual information through its vision-language alignment mechanism. Fusing these embeddings provides a richer video content representation, allowing the classifier to make more informed predictions. The integration of BLIP embeddings with the 3D CNN features leads to a significant improvement in accuracy. Specifically, there is an 18.05% increase (Table 2-A, row 1 - row 14) in accuracy for BLIP embeddings when combined with the Slow model, while the Slow model benefits from an improvement of 3.47% (Table 2-A, row 5, row 14). This trend underscores the advantage of utilizing *multimodal embeddings* that encapsulate textual and visual semantics. The second-best performing combination comprises the I3D model coupled with BLIP alignment embeddings. This pairing results in an accuracy improvement of 4.23% (Table 2-A, row 6 - row 13) and 17.38% (Table 2-A, row 1, row 13) for the I3D model and the BLIP embeddings, respectively. Similarly, the ResNet-3D model, when fused with cross-attention embeddings from BLIP, exhibits an increase of accuracy of 16.15% (Table 2-A, row 4, row 12) for UCF101. The highest accuracy achieved on the HMDB51 dataset

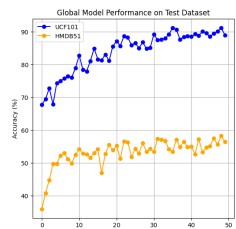

Figure 2: Accuracy achieved by the global model (*FLAMeST*) over 50 rounds (cycles).

is significantly lower compared to UCF101, which can be attributed to the inherent complexity and challenging nature of HMDB51.

Even in a centralized learning setup, the best performance achieved was approximately 71% (Table 3-B, row 15 ). In the FL setting, the highest accuracy of 67% (Table 2-B, row 14) was attained using the BLIP model and the Slow architecture. Our method achieves an improvement of 4.58% (Table 2-B, row 3, row 14) against the baseline. The BLIP and Slow models individually contributed to performance gains of 11.63% (Table 2-B, row 1, row 14) and 2.59% (Table 2-B, row 5, row 14), respectively, over their standalone performance. Furthermore, when BLIP embeddings were fused with the I3D model, an improvement of 4.89% (Table 2-B, row 6, row 13) in accuracy was observed, while the integration with ResNet-3D resulted in a 20.91% (Table 2-B, row 4, row 12) performance gain for the 3D CNN models.

*These findings suggest that incorporating semantic-visual embeddings alongside spatial-temporal embeddings enhances model performance by enriching feature representations with complementary contextual information.* Both the VLM and the CNN model benefit from the collaboration. Regarding the training accuracy of *FLAMeST* (Figure 2), the training accuracy improves and stabilizes with the subsequent communication cycle. We also calculate the cross-validation accuracy on UCF101 and HMDB51 over 20 folds across 80 rounds to get the average estimate over different train-test splits. The mean accuracy for UCF101 was approximately 94% with a standard deviation of 0.32 (Table 2-A, row 15), whereas for HMDB51, the mean calculated was 70% with a standard deviation of 0.98 (Table 2-B, row 15).

The InternVideo2.5 framework attains superior recognition performance, yielding an accuracy improvement of 1.32% over the proposed *BLIP + Slow* embeddings on the UCF101 dataset (Table 2–A, rows 16 and 14) and 3% on the HMDB51 dataset (Table 2–B, rows 16 and 15). The excellent quality of InternVideo2.5 embeddings is expected given its scale as the variant evaluated is 8B parameters, trained using a mixture of visual-text and long-video data (e.g., 3.7 M video-text pairs, 3.5 M image descriptions, and 0.7 M long videos). In contrast, our caption-based embedding backbone (BLIP) operates at an order of magnitude smaller size (hundreds of millions of parameters) and trained on fewer image-caption pairs.

Though the quality of embeddings is very good with InternVideo2.5, it imposes stringent hardware requirements, thereby constraining its feasibility for deployment on platforms with limited computational or memory resources.

In addition to action recognition, we evaluated the effectiveness of the BLIP cross-modal embeddings for image classification on the CIFAR-10 dataset. Our experiments achieved an accuracy of 90% after 30 communication rounds. Details are provided in Supplementary Section E.

### 5.3 Poor Performance of Static Text Embeddings Generated by CLIP

Unlike BLIP, which can generate captions directly for a given image, CLIP operates in a retrieval-based manner. To facilitate this process, we first generate a set of action-specific captions for each action in the UCF101 and HMDB51 datasets using large-scale language models such as ChatGPT. Each caption is then encoded using the text encoder of the CLIP model, and the resulting text embeddings are stored in a dictionary for efficient retrieval during training and inference (More details in the Supplementary A.3). Though computing embeddings in prior reduces the computational cost due to their static nature, the quality of embeddings is observed to be poor. As shown

Table 2: Federated Learning: Accuracy Comparison of Different Feature Extractors on UCF101 and HMDB51 Datasets. The bold-faced text denotes baseline and above accuracies. The baseline and *FLAMeST* values are indicated by * and ** prefixes, respectively.

(a) UCF101 Dataset

| S.No | Feature Extractor | Accuracy (%) |
|---|---|---|
| 1 | BLIP | 76.38 |
| 2 | CLIP | 60.98 |
| 3 | **KD (Baseline)** | ***89.30** |
| 4 | ResNet-3D | 64.01 |
| 5 | Slow | **90.96** |
| 6 | I3D | **89.53** |
| 7 | Video MAE | 65.04 |
| 8 | BLIP (Text Only) | 30.28 |
| 9 | BLIP (Text Only) + ResNet-3D | 64.00 |
| 10 | BLIP (Text Only) + Slow | **91.03** |
| 11 | BLIP (Text Only) + I3D | **89.92** |
| 12 | BLIP + ResNet-3D | 80.16 |
| 13 | BLIP + I3D | **93.76** |
| 14 | **BLIP + Slow** (*FLAMeST*) | ****94.43** |
| 15 | *FLAMeST* (**Cross Validation**) | **94±0.32** |
| 16 | InternVideo2.5 | **95.75** |

(b) HMDB51 Dataset

| S.No | Feature Extractor | Accuracy (%) |
|---|---|---|
| 1 | BLIP | 54.75 |
| 2 | CLIP | 30.98 |
| 3 | **KD (Baseline)** | ***61.80** |
| 4 | ResNet-3D | 38.95 |
| 5 | Slow | ***63.79** |
| 6 | I3D | 59.62 |
| 7 | Video MAE | 33.45 |
| 8 | BLIP (Text Only) | 27.18 |
| 9 | BLIP (Text Only) + ResNet-3D | 40.04 |
| 10 | BLIP (Text Only) + Slow | **63.19** |
| 11 | BLIP (Text Only) + I3D | 60.13 |
| 12 | BLIP + ResNet-3D | 59.86 |
| 13 | BLIP + I3D | **64.51** |
| 14 | **BLIP + Slow** (*FLAMeST*) | ****66.38** |
| 15 | *FLAMeST* (**Cross Validation**) | **70±0.98** |
| 16 | **InternVideo2.5** | **73.45** |

in Table 2-A, for UCF101, the accuracy obtained is 61% (Table 2-A, row 2), and for HMDB51, it is 33% (Table 2-B, row 2). Since these captions are generated before training, they are inherently generic and may fail to reflect individual video frames' unique visual and contextual attributes. *As a result, the fixed textual representation may not align well with the dynamic and heterogeneous nature of the visual embeddings, leading to suboptimal multimodal fusion and reduced classification accuracy.*

As the CLIP-generated embedding was not well refined, we did not conduct further studies on it, as most of the improvement would have come from the embeddings of the CNN model rather than the VLM in this joint learning.

## 5.4 Significance of Textual Embeddings

An additional set of experiments was conducted using only the textual embeddings generated by the VLM. The MLP model, in this case, was only trained on the text embeddings obtained from the VLM. Our findings resonate with recent analyses of embedding-based retrieval limits (Weller et al., 2025). They highlight weaknesses of single-vector representations, consistent with our observation that BLIP-only text embeddings are weak. For the BLIP model, when only text embeddings (captions generated) are used to train the MLP model, the accuracies obtained for UCF101 and HMDB51 are only 30.28% (Table 2-A, row 8) and 27.18% (Table 2-B, row 8), respectively. Aligning the text embeddings with the visual embeddings of the BLIP model improved accuracy by more than 27.57% for the HMDB51 dataset (Table 2-B, row 8 to row 1) and by 46.1% (Table 2-A, row 8 to row 1) for the UCF101 dataset. From this observation, we conclude that textual information alone cannot fully capture the dynamic nature of actions in video sequences. *While textual descriptions can introduce supplementary contextual details, they do not encompass the full range of spatial and temporal dependencies in videos.* However, textual information when used with visual features can

Table 3: Centralized Learning: Accuracy Comparison of Different Feature Extractors on UCF101 and HMDB51 Datasets. The bold-faced text denotes baseline and above accuracies. The baseline and *FLAMeST* values are indicated by * and ** prefixes, respectively.

(a) UCF101 Dataset

| S.No | Feature Extractor | Accuracy (%) |
|---|---|---|
| 1 | BLIP | 85.01 |
| 2 | CLIP | 62.80 |
| 3 | **KD (Baseline)** | **\*91.10** |
| 4 | ResNet-3D | 78.05 |
| 5 | Slow | **91.37** |
| 6 | I3D | **92.71** |
| 7 | Video MAE | 66.38 |
| 8 | BLIP (Text Only) | 34.73 |
| 9 | BLIP (Text Only) + ResNet-3D | 79.30 |
| 10 | BLIP (Text Only) + Slow | 90.67 |
| 11 | BLIP (Text Only) + I3D | 88.78 |
| 12 | BLIP + ResNet-3D | 85.82 |
| 13 | BLIP + I3D | **93.97** |
| 14 | ActionCLIP | **95** |
| 15 | **BLIP + Slow** (*FLAMeST*) | **\*\*94.30** |
| 16 | InternVideo2.5 | **96.75** |

(b) HMDB51 Dataset

| S.No | Feature Extractor | Accuracy (%) |
|---|---|---|
| 1 | BLIP | 62.36 |
| 2 | CLIP | 55.87 |
| 3 | **KD (Baseline)** | **64.10** |
| 4 | ResNet-3D | 53.87 |
| 5 | Slow | **63.17** |
| 6 | I3D | **65.24** |
| 7 | Video MAE | 39.38 |
| 8 | BLIP (Text Only) | 35.50 |
| 9 | BLIP (Text Only) + ResNet-3D | 52.92 |
| 10 | BLIP (Text Only) + Slow | **66.64** |
| 11 | BLIP (Text Only) + I3D | **65.98** |
| 12 | BLIP + ResNet-3D | **64.21** |
| 13 | BLIP + I3D | **68.04** |
| 14 | ActionCLIP | **76** |
| 15 | **BLIP + Slow** (*FLAMeST*) | **\*\*71.94** |
| 16 | InternVideo2.5 | **74.20** |

enhance the overall representative quality (Tables 2-(A, B), row 14). An illustration of the text captions generated by BLIP are shown in Supplementary C.2.

## 5.5 Comparison with Centralized Training

Table 3-(A, B) reports results under a centralized training setting, where all data is aggregated at a single site, effectively eliminating the challenges posed by data decentralization in FL. In this scenario, a single client possesses the entire dataset, and the MLP model is trained for 80 epochs using the same optimizer and learning rate as in the FL setup (Section 4). *As expected, centralized training consistently outperforms federated learning across most feature extractor combinations for a single cycle (Tables 2 and 3)*. This is primarily because, in FL, the model undergoes incremental updates over multiple communication rounds rather than maturing in a single training cycle. Consequently, in FL, the learning process is more gradual, and convergence takes longer compared to a centralized setting where the model has access to the complete dataset at all times. *Evaluating models in a centralized setup provides a valuable baseline for assessing the effectiveness of different feature extractors in a non-FL environment.* The results help determine whether a feature extractor is inherently strong or is hindered by federated constraints such as non-IID data distribution and communication limitations. The highest accuracy is achieved by the Internvideo2.5 model whereas the highest accuracy obtained with BLIP variation is 94.30% (Table 3-A, row 15) by the Slow and BLIP model when taken in conjunction with UCF101 and 71% (Table 3-B, row 15) for the HMDB51 dataset.

Table 4: Comparison of ActionCLIP and *FLAMeST* across different datasets and performance metrics in *Centralized Non-FL Setting.*

| Dataset | Method | Accuracy (%)↑ | Train Time (sec)↓ | Test Time (sec)↓ | Trainable Parameters↓ |
|---------|--------|---------------|-------------------|------------------|-----------------------|
| **UCF101** | ActionClip | **95** | 242.993 | 96.51 | 150-155 million |
|  | *FLAMeST* | 93.34 | **5.862** | **0.595** | 1 million |
| **HMDB51** | ActionClip | **75.89** | 159.170 | 15.001 | 150-155 million |
|  | *FLAMeST* | 73.36 | **3.637** | **0.379** | 1 million |

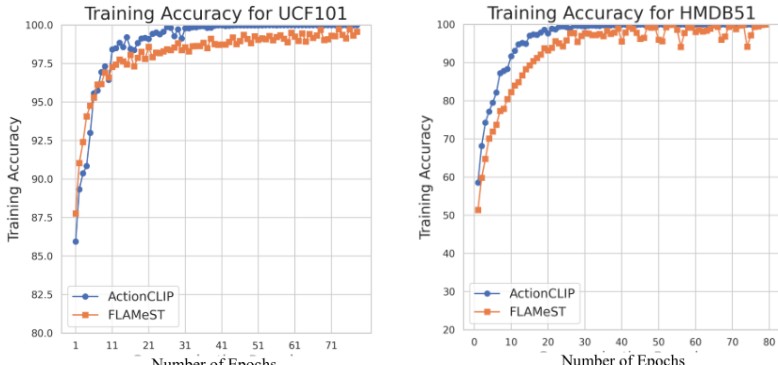

Figure 3: Centralized training accuracy trends of ActionCLIP and *FLAMeST* over 80 epochs. The graph illustrates how the models perform across training, highlighting the stability and convergence behavior of each method.

### 5.6 Comparison of ActionCLIP and *FLAMeST*

We trained the ActionCLIP[9] model from scratch, utilizing its default parameter settings, for a total of 80 epochs on our specific data partition (Section4). The results of this training are presented in Table 4, which provides a comparative analysis between *FLAMeST* and ActionCLIP. The comparison reveals that ActionCLIP outperforms *FLAMeST* by achieving a 2% higher accuracy on both UCF101 and HMDB51. However, *FLAMeST* demonstrates notable advantages in terms of efficiency, with significantly reduced training and inference times compared to ActionCLIP. Furthermore, *FLAMeST* operates with a substantially lower number of training parameters, making it a more lightweight and computationally efficient option relative to ActionCLIP. Figure 3 shows that over the epochs, *FLAMeST* also acquires comparable training accuracy to ActionCLIP, leading to the model convergence.

### 5.7 Understanding Embedding Quality through UMAP Projections

To assess the quality of the embeddings generated by the BLIP and Slow 3D CNN, we utilize UMAP—a widely used dimensionality reduction and visualization technique for high-dimensional data (McInnes et al., 2018). For this analysis, we randomly select two classes from each dataset,

---

[9]https://github.com/sallymmx/ActionCLIP

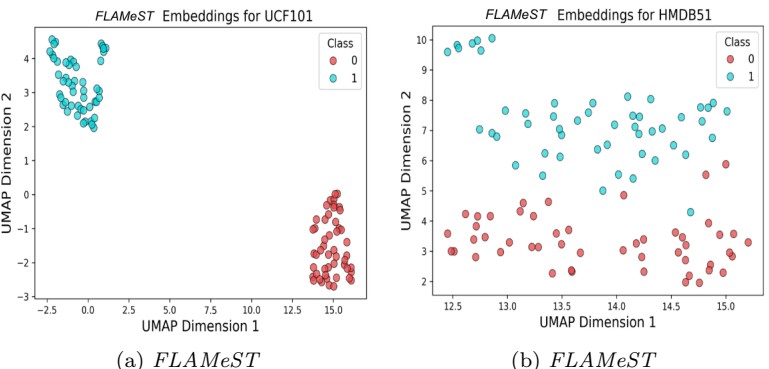

Figure 4: UMAP visualization of different embedding strategies on selected classes from the UCF101 and HMDB51 datasets. For UCF101, we consider the classes 'Apply Lipstick' and 'Archery'. For HMDB51, the selected classes are 'Catch' and 'Cartwheel.'

UCF101 and HMDB51, and visualize the corresponding embeddings in a two-dimensional space. As illustrated in Figure 4 (a,b), the *FLAMeST* embeddings for UCF101 and HMDB51 form *well-separated and compact clusters, indicating that the combined representation effectively distinguishes between the two selected classes, demonstrating the discriminative power of the embeddings.* More UMAP Figures A-3 are provided in Supplementary C.1 .

## 6 Ablation Studies

The experimental configuration for the ablation study, including the model architecture, hyperparameters, and data partitioning strategies, remains consistent with the setup outlined in Section 4. The results quoted are the best accuracy obtained by the global model over 80 rounds.

### 6.1 Client-Side Model Update

Table 5 evaluates federated optimization algorithms—FedAvg (McMahan et al., 2017), FedProx (Li et al., 2020) and FedDyn (Acar et al., 2021) under both IID and Non-IID data distributions. FedAvg (EQ. 11) is the baseline aggregation method, where client updates are averaged without accounting for data distribution differences. In contrast, FedProx introduces a proximal term in the local objective function to address client data heterogeneity (EQ. A-1). Whereas FedDyn uses a dynamic update of the local objective loss function to ensure consistency of the local model update with the global model update (EQ. A-2).

Table 5 shows that all three methods perform comparably under the IID setting. Whereas for the Non-IID case, FedProx outperforms FedAvg and FedDyn for the UCF101 and HMDB51 datasets. This improvement highlights the effectiveness of *FedProx in handling the statistical challenges posed by heterogeneous data environments, making it a more suitable choice in real-world scenarios where data is often non-uniformly distributed across clients.*

Table 5: Comparison of client update methods on IID and Non-IID settings across datasets (4 clients).

| Dataset | Method | IID (%) | Non-IID (%) |
|---------|--------|---------|-------------|
| **UCF101** | FedAvg | 90 | 84 |
| | FedProx | **93** | **86** |
| | FedDyn | 91 | 84 |
| **HMDB51** | FedAvg | 65 | 60 |
| | FedProx | *67* | **62** |
| | FedDyn | **69.15** | 60.81 |

Table 6: Performance of different MLP architectures on the UCF101 dataset.

| S.no | Hidden Layers | Train Acc. (%) | Test Acc. (%) |
|------|---------------|----------------|---------------|
| 1 | *[512, 256]* | *99.07* | *93.89* |
| 2 | [256, 256] | 98.50 | 94.30 |
| 3 | [1024, 512] | 99.37 | 96.22 |
| 4 | [1024, 512, 256] | 98.48 | 94.14 |
| 5 | [1024, 1024, 1024] | 98.80 | **96.30** |

## 6.2 Classifier Network Architecture

As shown in Table 6, increasing the width of hidden layers improves test performance, with the [1024, 1024, 1024](row 5) model achieving the highest accuracy of 96.3%. However, deeper networks do not always yield better results—[1024, 512, 256](row 4) underperforms compared to the simpler [1024, 512](row 3) model, suggesting that additional depth may introduce optimization difficulties or lead to diminishing returns. All architectures achieve high training accuracy (>98%), indicating minimal overfitting. *A moderately deep and wide MLP is often sufficient to achieve strong generalization, while excessively deep models may not offer significant gains.*

## 6.3 Effect of Epochs on Model Performance

Table 7 presents the impact of training epochs on model accuracy. For UCF101 under IID settings, accuracy begins at 96% with 5 epochs and stabilizes near 94.8% by epoch 15, indicating early convergence. In contrast, Non-IID performance fluctuates. It rises from 84% to a peak of 86% at epoch 10, then declines to 83%, suggesting potential overfitting under data heterogeneity. On HMDB51, the IID model improves consistently from 70% to 73%, whereas the Non-IID model shows modest gains from 58% to 61%, further highlighting the challenges of learning from skewed distributions

*While increasing the number of epochs helps improve or stabilize performance under IID settings, the same trend does not hold under Non-IID conditions.*

## 6.4 Scalability with Increasing Clients

For Scalability Approaches 1A, 1B, and 2, the total dataset size remains constant, so as the number of clients increases, the amount of data per client decreases. In contrast, for Approaches 3A and 3B, the data assigned to each client remains fixed, meaning the per-client dataset size does not decrease when the number of clients increases.

### 6.4.1 Scaling Approach: 1A & 1B

Table 8 presents the performance of *FLAMeST* as the number of clients increases in a cross-silo FL setting, where each client typically represents an institution. We limit the client count to 10 to reflect realistic deployment scenarios. The data is divided among the clients unequally, i.e., one client may get more data and the other may get much less. Experiment 1A corresponds to an

IID distribution, and Experiment 1B corresponds to a non-IID distribution among the clients. As the number of clients increases, classification accuracy declines for both the UCF101 and HMDB51 datasets in the IID (rows 1 and 2) and Non-IID scenarios (rows 3 and 4). *The degradation is primarily due to reduced data per client as the number of clients increases, which weakens local training and limits global model generalization.*

### 6.4.2 Scaling Approach: 2

We investigate label-skewness scaling under the constraint that the per-class-per-client dataset sizes remain fixed. Each client is first assigned a balanced base dataset containing an equal number of samples from every class. Class-specific heterogeneity is then introduced through a probabilistic allocation mechanism (Supplementary Section A.4). Specifically, class proportions are drawn from a Beta distribution to determine the relative representation of each class, while a Bernoulli distribution identifies the subset of clients (takers) associated with that class. Once the participating clients for a given class are selected, the corresponding sample quotas are evenly distributed among them, thereby ensuring fairness in allocation while preserving skewness in label distribution. As observed in Table 8 (rows 5 and 6), an increase in the number of clients leads to a gradual decline in classification accuracy for both UCF101 and HMDB51, primarily due to the heightened data heterogeneity and decrease in the data per client. *Interestingly, for HMDB51, scaling results in an accuracy improvement that even surpasses the performance achieved under the IID setting.*

### 6.4.3 Scaling Approach: 3A & 3B

To assess the scalability of federated learning (FL) under realistic distributed conditions, we perform a *single* Dirichlet partition of the dataset with concentration parameter $\beta = 0.6$ (Non-IIID) and $\beta = 10$ (IID) into ten clients. This partition is created once and reused across all experiments (Supplementary Section A.5). Each client permanently retains its assigned local dataset, ensuring that the **per-client data remains fixed** irrespective of the number of participating clients. No data reshuffling or redistribution is performed when varying the number of clients participating in FL. Initially, 4 randomly selected clients participate in FL, and we compute the metrics. The next 8 clients participate in FL, including the 4 previously selected ones and 4 newly randomly selected; however, all models are reinitialized from scratch. The next 10 clients participate in FL, including the 8 previously selected ones and 2 newly randomly selected; however, all models are reinitialized from scratch.

The nested design i.e. incremental client addition is essential to maintain experimental consistency as it guarantees that any observed performance variation arises solely from the increase in the number of participating clients, rather than from changes in data composition. Experiments 3A and 3B correspond to IID and Non-IID distributions, respectively. As shown in the Table 8 (rows 7,8,9,10), the average improves as the number of participating clients increases. For IID there is an increases from **89.18%** (4 clients) to **91.40%** (10 clients) for UCF101 and **59.70%** (4 clients) to **64.68%** (10 clients) for HMDB51 respectively (row 7,8) The average global accuracy in the case of Non-IID also increases from **82.91%** (4 clients) to **87.63%** (8 clients) and **88.04%** (10 clients) for UCF101 (row 9) and **49.67%** (4 clients) to **59.68%** (10 clients) for HMDB51 (row 10).

This trend indicates that adding more clients contributes to broader data coverage and richer inter-client diversity, allowing the global model to generalize better. A similar pattern is observed across different random seeds, where increasing the federation size enhances model performance.

Table 7: Accuracy (%) across varying training epochs on UCF101 and HMDB51 over 80 cycles for 4 clients.

| S.no | Epochs | UCF101 IID | UCF101 Non-IID | HMDB51 IID | HMDB51 Non-IID |
|------|--------|------------|----------------|------------|----------------|
| 1 | 5 | **96.0** | **84.0** | 70.0 | 58.0 |
| 2 | 10 | 94.7 | 86.0 | 71.0 | 59.0 |
| 3 | 15 | 94.8 | 83.0 | **73.0** | **61.0** |

Table 8: Accuracy (%) across varying client counts on UCF101 and HMDB51 over 80 cycles with 5 local epochs. Rows are partitioned based on scalability experiments 1A, 1B, 2, 3A, and 3B.

| S.no | Setting | 4 Clients | 8 Clients | 10 Clients |
|------|---------|-----------|-----------|------------|
| | **Scalability 1A** | | | |
| 1 | UCF101 (IID) | 96 | 93 | 91 |
| 2 | HMDB51 (IID) | 67 | 63 | 57 |
| | **Scalability 1B** | | | |
| 3 | UCF101 (Non-IID) | 86 | 82 | 77 |
| 4 | HMDB51 (Non-IID) | 55 | 47 | 37 |
| | **Scalability 2 — Fixed Class per Client** | | | |
| 5 | UCF101 | 84.93 | 83.93 | 82.49 |
| 6 | HMDB51 | 74 | 72.96 | 72 |
| | **Scalability 3A — Fixed Dataset per Client (IID)** | | | |
| 7 | UCF101 (IID) | 89.18 | 90.74 | **91.40** |
| 8 | HMDB51 (IID) | 59.70 | 63.30 | **64.68** |
| | **Scalability 3B — Fixed Dataset per Client (Non-IID)** | | | |
| 9 | UCF101 (Non-IID) | 82.91 | 87.63 | **88.04** |
| 10 | HMDB51 (Non-IID) | 49.67 | 58.44 | **59.68** |

Overall, these results confirm that the proposed FL framework effectively scales with distributed data growth, demonstrating stable convergence and improved accuracy as the federation expands.

## 6.5 Fusion by Gated-Attention

In order to study the effectiveness of a more advanced fusion technique than plain concatenation, we have experimented with *residual gated cross fusion* (RGCF) between image and text embeddings (Refer to Supplementary section A.6). Table 9 shows that simple concatenation consistently outperforms RGCF on both UCF101 and HMDB51, even when RGCF is trained for more epochs. However, the accuracy of RGCF does improve with increased training—from 75% to 89.5% on UCF101 and from 56% to 58.8% on HMDB51, indicating that RGCF may benefit from longer training to stabilize. We suspect the average performance of RGCF is due to the added complexity and parameterization of RGCF, leading to overfitting on limited local data. In contrast, simple concatenation preserves the full representational capacity of CNN and VLM embeddings without additional transformations.

Table 9: Performance comparison of Residual Gated Cross Fusion (RGCF) and simple concatenation method on UCF101 and HMDB51 datasets over 80 cycles and 4 clients.

| S.no | Dataset | RGCF (5 epochs) | RGCF (50 epochs) | Simple Concatenation |
|------|---------|-----------------|------------------|----------------------|
| 1 | UCF101 | 75 | 89.5 | **94.43** |
| 2 | HMDB51 | 56 | 58.8 | **70** |

## 7 Failure Analysis

We observe that broadly, the error cases fall into three main categories: (A) selection of uninformative frames, (B) high inter-class similarity, and (C) noisy or incomplete caption VLM. Detailed discussion of failure cases is presented in (Supplementary D) and their mitigation strategies are discussed in (Supplementary D.1).

- **Uninformative Frame Selection:** In *FLAMeST*, frames are sampled randomly from the video clips. Consequently, the selected frame may not adequately capture the action being performed. The frames corresponding to the "kayaking" and "haircut" classes fail to depict critical visual cues such as a kayak or scissors—objects that are central to recognizing the activity (Figures A-5-A and A-6-A).

- **High Inter-Class Similarity:** Certain action classes exhibit substantial visual overlap, particularly when temporal context is omitted. For example, "kayaking" and "rafting" both involve similar water-based environments and the presence of boats, making static frame-based distinction challenging (Figures A-5-B and A-6-B).

- **Noisy or Incomplete Captions:** The VLM's inability to generate accurate and descriptive captions for the selected frames (Supplementary Figures A-5-C and A-6-C) along with the qualitative analysis of the captions generated by VLM shown in Supplementary Section B suggests that captions alone are often insufficient for complex actions as text can be noisy. Our fusion with Slow CNN embeddings mitigates this, but reliance on captions remains a limitation.

## 8 Broader Applicability

Although our experimental study focuses on video action recognition on UCF101 and HMDB51, the FLAMeST design is more general. The key idea of using frozen multimodal backbones to compute local embeddings and federating only a lightweight prediction head, matches recent trends in parameter-efficient federated learning with foundation models. In principle, the same template can be instantiated for other video and multimodal problems where a task can be formulated as a mapping from pre-trained embeddings to outputs, such as multi-label event detection, gesture recognition, video retrieval, egocentric activity understanding, or audio–visual classification. Similarly, the BLIP + 3D CNN backbone can be replaced by stronger or task-specific encoders (e.g., alternative VLMs or video models like InternVideo-style architectures), and even heterogeneous backbones across clients, as long as their representations are projected into a compatible feature space. In all these cases, federating only a small head (or other lightweight adaptation modules) keeps communication and compute costs manageable while allowing the framework to benefit from more diverse datasets and model configurations.

## 9 Conclusions and Future Work

This study introduces *FLAMeST*, an approach for integrating foundation models within an FL framework to enhance VAR performance. We leverage embeddings from the BLIP model alongside features extracted by a 3D CNN model called Slow. These combined representations train a lightweight MLP in the federated cycle, substantially reducing communication overhead compared to transmitting the full foundation model. The fusion of semantic and visual embeddings yields notable accuracy gains on challenging benchmarks such as HMDB51 and UCF101. Ablation studies confirm *FLAMeST's* robustness across diverse client-update schemes and its scalability under both IID and Non-IID data distributions. Moreover, the *FLAMeST* embeddings form well-clustered class representations, highlighting their discriminative richness. Comparative analyses against alternative strategies further demonstrate how *FLAMeST* effectively addresses key challenges in VAR tasks in a collaborative set-up.

As part of future work, we intend to investigate distilled variants of these foundation models to reduce storage and computation costs, thereby improving the practicality of edge-device FL. Another direction involves enabling client-side customization, wherein users can record and label video clips. This introduces new challenges around server-side training and privacy preservation, which merit deeper investigation. As future work, we plan to explore distilled or lightweight video-language captioners that could be feasible for deployment in federated learning without compromising privacy.

## Acknowledgement

The authors acknowledge support from the Department of Computer Science and Engineering for providing access to a high-performance computing facility based on NVIDIA DGX equipment. The authors also thank the Government of India (GoI) ANRF/SERB project CRG/2023/007282 for partial financial support.

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
