

Figure A-1: In FLAMeST, only the MLP component of each client's model participates in the federated learning process. (A) Each client possesses its own dataset. The foundation models remain frozen and are used to generate embeddings from the client data, which are then used to train the client's MLP model. (B) After local training, each client uploads its MLP model to the central server. (C) The server aggregates these MLP models by averaging their parameters to create a global MLP model. (D) This global MLP model is then sent back to the clients. (E) Each client downloads and integrates the updated global MLP model for further training or inference.

## A    Supplementary Material

Section A.1 provides a detailed description of the k-Means-based keyframe selection process, which also accounts for the temporal order of frames to effectively preserve motion information. A detailed pictorial representation of the FL setup for *FLAMeST* is shown in Figure A-1. The detailed description of the client optimization technique - FedProx (Li et al., 2020) and FedDyn (Acar et al., 2021) is discussed in Section A.2. The training and testing procedures for the CLIP-based feature extraction model are elaborated in Section A.3. Section A.6 introduces the Residual Gated Cross Fusion mechanism, which leverages cross-attention between image and text embeddings for the fusion process. A quantitative analysis is done for the text embeddings, which is presented in Section B. A qualitative evaluation of FLAMeST is presented in Section C, which includes UMAP visualizations of the BLIP and Slow model embeddings (Section C.1) as well as an analysis of the caption quality generated by the BLIP model (Section C.2). We further examine the failure

Algorithm A-1: Algorithm - Order Preserving Frame Clustering (OPFC)

1: Input $X = [x_1, \ldots, x_N]$ (video clip), $K$ (given hyperparameter)
2: Initialize centers randomly, $(\forall i \in [1, \ldots, K]) : \xi_i \in R^{(3,W,H)}$
3: $M = \{(i,j)|(\forall i \in [1, \ldots, N], \exists j \in [1, \ldots, k])\}$ (randomly assign frame to clusters)
4: Perform k-Means for I iterations, keeping track of member frame indices
5: **for** $i \in [1, \ldots, I]$ (for $I$ iterations) **do**
6:    **for** $j \in [1, \ldots, N]$ **do**
7:       $k^* = \underset{k \in [1,\ldots,k]}{argmin}\textbf{DISTANCE}(x_j, \xi_k)$
8:       Let $\exists (j, j^*) \in M$ (determine previously mapped cluster)
9:       $M = M - \{(j, j^*)\}$
10:      $M = M \cup \{(j, k^*)\}$
11:    **end for**
12:    (Re-compute centroid for each cluster)
13:    $(\forall k \in [1, \ldots, K]) : \xi_k = \textbf{CENTROID}(\{x_j|(j,k) \in M\}$
14: **end for**
15: $\Gamma = []$ (Determine earliest instances in each cluster)
16: **for** $k \in [1, \ldots, K]$ **do**
17:    $\Gamma = \Gamma \bigodot min\{a|(a,b) \in M, b = k\}$
18: **end for**
19: Sort in chronological order of frame events and return clustered clip
20: **return** $\{x_j|\forall j \in \textbf{SORT}(\Gamma)\}$

cases of FLAMeST in Section D, followed by a brief discussion of possible mitigation strategies in Section D.1.

## A.1 Order Preserving Frame Clustering (OPFC)

In general, there may be repeating information between successive image frames in several of the actions (Figure A-2). The algorithm (Algorithm A-1) is a modification on top of standard *k-Means* (Lloyd, 1982) for handling membership of frame indices. The *CENTROID(·)* function takes as input a set of image frame tensors and computes their average. The *SORT(·)* function performs sorting of the indices in ascending order.

### Determining the Value of "k" for Clustering

We use a subset of the UCF101 and Hmdb51 datasets for determining the optimal value of "k". We experimented with k $\in \{2, 3, 5, 8\}$ and based on the elbow method, we identified $k = 3$ as the optimal choice. In this selection process of "k", we also considered both the silhouette score and a confidence score. The rationale behind incorporating the confidence score is to identify an optimal k where the confidence interval is low, indicating minimal cluster variance, which improves clustering quality. To make clustering more efficient and effective, we downscale the original frame resolution from the original dimension to $64 \times 64$ before applying K-means clustering. To retain the full context, we use the *actual frames* after clustering the indices.

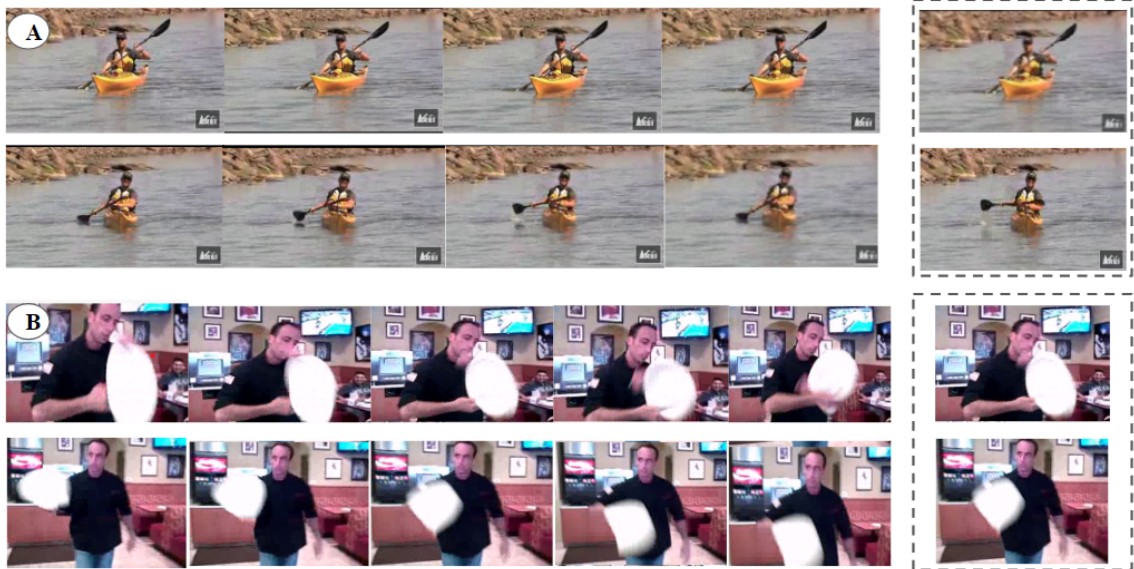

Figure A-2: Images on the left side show a snapshot of some frames taken from a cluster for the category *(a) Kayaking and (b) Pizza tossing.* The highlighted image on the right is a representative image of each cluster. The two rows represent two different clusters obtained using (Algorithm A-1)

## A.2 Client Update

If the client model is updated according to Fedprox (Li et al., 2020), then along with minimizing the cross-entropy loss for each client $i$, it also minimizes the $L_2$ norm between the local model and the global model $\mathbf{w}_t^*$ [10]. The hyperparameter $\mu$ controls the balance between the two components of the loss function ( A-1). A higher value of $\mu$ encourages local models to align more closely with the global model updates, promoting consistency across the federated network. Conversely, a lower $\mu$ value allows greater flexibility in local model updates.

$$\mathcal{J}(\mathbf{w}) = \frac{1}{|\mathcal{D}|} \sum_{(e,y)\in\mathcal{D}} \mathcal{L}_{\mathrm{CE}}(g(e), y) + \frac{\mu}{2} \left\| \mathbf{w} - \mathbf{w}_{(t^*)} \right\|_F^2 \tag{A-1}$$

$$\mathcal{J}(\mathbf{w}) = \frac{1}{|\mathcal{D}|} \sum_{(e,y)\in\mathcal{D}} \mathcal{L}_{\mathrm{CE}}(g(e), y) - \left\langle \mathbf{h}_{(t)}, \mathbf{w} \right\rangle + \frac{\lambda}{2} \left\| \mathbf{w} \right\|_F^2 \tag{A-2}$$

The client can also update according to the FedDyn (Acar et al., 2021) objective  A-2. Here, $\mathcal{D}$ denotes the local client dataset, and $\mathcal{L}_{\mathrm{CE}}$ is the standard cross-entropy loss applied to the model output $g(e)$ and the true label $y$. The term $\left\langle \mathbf{h}_{(t)}, \mathbf{w} \right\rangle$ represents a linear correction based on the global history vector $\mathbf{h}_{(t)}$, which helps align local updates with the global optimization trajectory.

---

[10] $\mathcal{J}^i$ denotes loss for $i^{th}$ client

The final term, $\frac{\lambda}{2}\|\mathbf{w}\|_F^2$, serves as an $\ell_2$ regularization to control the norm of the local model parameters.

Each client has a pre-trained VLM and a CNN model, both frozen during training and inference. **The only trainable component is the MLP**, which participates in a FL cycle (Algorithm 2).

### A.3  CLIP-Based Feature Extraction for Pre-Computed Text Embeddings.

In addition to BLIP, we utilize the CLIP model as another vision-language representation learning approach. Unlike BLIP, which can generate captions directly for a given image, CLIP operates in a retrieval-based manner. Given an input frame, CLIP selects the most relevant caption from a predefined set, making its data transformation process distinct from that of BLIP. To facilitate this process, we first generate a set of action-specific captions for each action in the UCF101 and HMDB51 datasets using large-scale language models such as ChatGPT. Each caption is then encoded using the text encoder of the CLIP model, and the resulting text embeddings are stored in a dictionary for efficient retrieval during training and inference.

### A.3.1  Training Phase

A candidate frame is passed through the CLIP image encoder during training to extract its visual embeddings. Since the ground-truth action label is available during training, the corresponding text embedding is retrieved from the precomputed dictionary. The retrieved text embedding and the extracted visual embedding are then concatenated to form the final multimodal representation, which is subsequently used for training the MLP model.

### A.3.2  Testing Phase

A different data transformation procedure is employed during inference as the ground-truth action labels are unavailable. Similar to the training process, the image encoder of CLIP extracts visual embeddings for the candidate frame. However, since the corresponding text embedding cannot be directly retrieved from the dictionary due to the absence of the ground truth, a similarity-based approach is used instead. Specifically, cosine similarity is computed between the visual and the list of precomputed text embeddings. The text embedding with the highest similarity score is selected as the most relevant representation for the given frame. The retrieved textual and corresponding image embeddings are then fed to the MLP model for inference.

### A.4  Scaling in a Non-IID Setup: Approach 2

We design the dataset distribution procedure under a label-skewed non-IID setup. The construction of client datasets proceeds in two stages: a *base allocation* stage and a *heterogeneous allocation* stage.

**Base Allocation.**  Each client is first assigned a balanced base dataset. Specifically, an equal number of samples from each class are drawn, as formalized in Eq. equation A-3, and these are distributed uniformly across clients as in Eq. equation A-5. This guarantees that all clients begin with the same balanced core, thereby avoiding degenerate cases where a client has no exposure to certain classes.

**Heterogeneous Allocation.** On top of the base dataset, heterogeneity is introduced by probabilistically assigning clients to specific classes:

1. **Class Probability Sampling.** For each class $i$, a probability $p_i$ is sampled from a Beta distribution (EQ. A-6). This probability controls how likely it is that a client will be assigned to class $i$.

2. **Client Taker Selection.** Each client $h$ is then selected as a taker for class $i$ according to an independent Bernoulli trial with probability $p_i$ (EQ. A-7). The set of takers for class $i$ is denoted $A_i$ (EQ. A-8).

3. **Quota Assignment.** Once the set $A_i$ is determined, the total number of samples of class $i$ is evenly split across the takers. The quota per client for class $i$ is defined in EQ. A-9.

4. **Subset Allocation.** Each taker $h \in A_i$ receives a subset of $q_i$ samples from the class pool $T_i$, drawn uniformly without replacement, as described in EQ. A-10.

**Final Dataset.** The final dataset of client $h$ is then given by the union of its balanced base dataset and the heterogeneous allocations from the above procedure, as defined in EQ. A-11.

Classes $\mathcal{K} = \{1, \ldots, K\}$, clients $\mathcal{C}l = \{1, \ldots, C\}$.
Class-$i$ pool $T_i$ with $N_i = |T_i|$. Base per class $B$.

**Base (identical to all clients).**

$$B_i \sim \text{Unif}\{S \subseteq T_i : |S| = B\}, \tag{A-3}$$

$$B = \bigcup_{i=1}^{K} B_i, \tag{A-4}$$

$$D_h^{(0)} = B \quad \forall h \in \mathcal{C}l. \tag{A-5}$$

**Random taker subset per class (keep Beta).**

$$p_i \sim \text{Beta}(a, b), \tag{A-6}$$

The Beta distribution is used to model class heterogeneity across clients, as it naturally generates proportions within [0,1]. By tuning its parameters, it can simulate both balanced and highly skewed label distributions, making it well-suited for creating realistic non-IID scenarios in federated learning.

$$Z_{i,h} \mid p_i \overset{\text{i.i.d.}}{\sim} \text{Bernoulli}(p_i), \tag{A-7}$$

$$A_i = \{h \in \mathcal{C}l : Z_{i,h} = 1\}, \qquad M_i = |A_i|. \tag{A-8}$$

If $M_i \geq 1$, set

$$q_i = \left\lfloor \frac{N_i}{M_i} \right\rfloor. \tag{A-9}$$

(If $q_i = 0$ we will randomly skip extras for class $i$).

For each $h \in A_i$, draw independently

$$S_{i,h} \sim \text{Unif}\{S \subseteq T_i : |S| = q_i\}. \tag{A-10}$$

(We do not remove base items; duplicates within a client can arise via base/extras).

**Final client datasets (multiset union).**

$$D_h = D_h^{(0)} \uplus \bigcup_{i:h \in A_i} S_{i,h}, \qquad h \in \mathcal{Cl}. \tag{A-11}$$

### A.5 Scaling in a Non-IID Setup (Fixed dataset per client): Approach 3A & 3B

**Objective.** To evaluate how the federated learning (FL) model performance scales as the number of clients increases, while keeping each client's local dataset fixed (even if clients have different local data sizes).

**Setup.** The complete training dataset is partitioned once into 10 clients, denoted as $\{\mathcal{C}_1, \mathcal{C}_2, \ldots, \mathcal{C}_{10}\}$. Each client retains its own local dataset permanently; no data reshuffling or reallocation occurs between experiments. Refer to Table 8 for data partitioning details.

**Nested Client Configurations.** To study scalability fairly, we form nested subsets of clients (S) so that each larger configuration strictly adds new clients:

$$\mathcal{S}_4 \subset \mathcal{S}_8 \subset \mathcal{S}_{10},$$

corresponding to federations with $K \in \{4, 8, 10\}$ total clients. This design ensures that any change in performance reflects the addition of new clients (and data), rather than changes in client composition.

**Training Protocol.** For each configuration ($K = 4, 8, 10$), we perform an independent FL training run with the following settings:

- **Algorithm:** FedAvg.

- **Model:** identical architecture and initialization across runs.

- **Communication rounds:** $T = 80$.

- **Local epochs:** $E = 5$ (fixed).

- **Participation:** all $K$ clients participate in the experiment.

- **Hyperparameters:** learning rate, optimizer, and batch size are kept identical.

- **Evaluation:** global model accuracy is measured on the same test set.

### A.6 Residual Gated Cross Fusion Formulation

Let $\boldsymbol{X}_I \in \mathbb{R}^{B \times d_I}$ denote the CNN-based image embeddings. $\boldsymbol{X}_T \in \mathbb{R}^{B \times d_T}$ and the text embeddings (e.g., from BLIP) where $B$ is the batch size and $d$ is the common projected embedding dimension.

**1. Projection to Common Space :**

$$\boldsymbol{X}'_I = \boldsymbol{X}_I \boldsymbol{W}_I + \boldsymbol{b}_I \tag{A-12}$$

$$\boldsymbol{X}'_T = \boldsymbol{X}_T \boldsymbol{W}_T + \boldsymbol{b}_T \tag{A-13}$$

where $\boldsymbol{W}_I \in \mathbb{R}^{d_I \times d}$, $\boldsymbol{W}_T \in \mathbb{R}^{d_T \times d}$.

**2. Cross Attention :** We compute how each modality attends to the other.

$$\text{Att}_{I \to T} = \text{softmax}\left(\frac{(\boldsymbol{X}'_I \boldsymbol{W}_Q)(\boldsymbol{X}'_T \boldsymbol{W}_K)^\top}{\sqrt{d}}\right)(\boldsymbol{X}'_T \boldsymbol{W}_V) \tag{A-14}$$

$$\text{Att}_{T \to I} = \text{softmax}\left(\frac{(\boldsymbol{X}'_T \boldsymbol{W}_Q)(\boldsymbol{X}'_I \boldsymbol{W}_K)^\top}{\sqrt{d}}\right)(\boldsymbol{X}'_I \boldsymbol{W}_V) \tag{A-15}$$

where $\boldsymbol{W}_Q, \boldsymbol{W}_K, \boldsymbol{W}_V \in \mathbb{R}^{d \times d}$.

**3. Residual Gated Fusion :** This controls how much of the attended vs. original embedding is retained.

$$\boldsymbol{g}_I = \sigma\left([\boldsymbol{X}'_I; \text{Att}_{T \to I}]\boldsymbol{W}_g^I\right) \tag{A-16}$$

$$\boldsymbol{g}_T = \sigma\left([\boldsymbol{X}'_T; \text{Att}_{I \to T}]\boldsymbol{W}_g^T\right) \tag{A-17}$$

where $\sigma$ is the sigmoid function, and $[\,;\,]$ denotes concatenation.

$$\boldsymbol{F}_I = \boldsymbol{g}_I \odot \text{Att}_{T \to I} + (1 - \boldsymbol{g}_I) \odot \boldsymbol{X}'_I \tag{A-18}$$

$$\boldsymbol{F}_T = \boldsymbol{g}_T \odot \text{Att}_{I \to T} + (1 - \boldsymbol{g}_T) \odot \boldsymbol{X}'_T \tag{A-19}$$

**4. Final Fusion Output :** We then concatenate the gated representations and pass them through a residual MLP.

$$z = \text{MLP}([\boldsymbol{F}_I; \boldsymbol{F}_T]) + [\boldsymbol{F}_I; \boldsymbol{F}_T] \tag{A-20}$$

## B  Quantitative Caption Analysis

To assess the semantic alignment between generated captions and the ground-truth action labels, we employ two complementary quantitative strategies. The first is an embedding-based similarity metric that evaluates alignment in the latent space of a pretrained text encoder, while the second is a lexical overlap strategy that directly checks for surface-level matches between caption tokens and label variants. Together, these methods provide a balanced evaluation of caption–label consistency at both semantic and lexical levels.

### B.1 Matching Embeddings

To quantitatively evaluate the semantic consistency between generated captions and the ground-truth labels, we design the following metric. This metric focuses on assessing the alignment between textual representations of captions and label variants by leveraging pretrained text encoders. The details are outlined below:

- Step 1: Encode the caption using a text BLIP-text encoder.

- Step 2: Encode each label variant (or short templates such as "a person {label}").

- Step 3: Compute the score:

$$s_{\text{text}} = \max_j \cos\left(f_{\text{text}}(\text{caption}), f_{\text{text}}(\text{label}_j)\right)$$

- Step 4: Select the top 1 and top 5 candidates obtained from Step 3.

- Step 5: Calculate the top 1 and top 5 accuracy.

**Using cosine similarity between caption embeddings and label embeddings, Top-1/5 accuracy was  7% and 15% for both the datasets.**

### B.2 Lexical Overlap Strategy

To evaluate the semantic consistency between generated captions and ground-truth labels, we adopt the following caption–label alignment verification procedure:

- Step 1: Caption Generation: A BLIP model is employed to generate descriptive captions for each image.

- Step 2: Tokenization: The generated caption is decomposed into a sequence of tokens for subsequent comparison.

- Step 3: Ground-Truth Matching: The corresponding ground-truth label is compared against the caption tokens (including possible variants). If a match is detected, a counter is incremented to record successful alignment.

- Step 4: Statistical Aggregation: The counts of matches and total comparisons are accumulated across the dataset to compute overall alignment statistics.

**Lexical overlap:  Only 6.1% of HMDB51 and 20.7% of UCF101 captions contained the action label (or split token).**   These results confirm that captions are weakly aligned with action semantics, motivating multimodal fusion.

## C   Qualitative Analysis

In this section, we do a visualization study using UMAP on the embeddings generated by *FLAMeST* and glance through the quality of the caption generated by the BLIP model.

### C.1 Understanding Embedding Quality through UMAP Projections

To assess the quality of the embeddings generated by the BLIP and Slow 3D CNN models, we utilize UMAP—a widely used dimensionality reduction and visualization technique for high-dimensional data (McInnes et al., 2018). For this analysis, we randomly select two classes from each dataset, UCF101 and HMDB51, and visualize the corresponding embeddings in a two-dimensional space. For the UCF101 and HMDB51 datasets, the cross-attention embeddings obtained using only the BLIP model also perform well (Figure A-3 (c,d)). Though some data points are lying closer to the decision boundaries, potentially leading to misclassification, the embeddings are overall tightly grouped. When visualized using UMAP, the embeddings extracted from the Slow 3D CNN (Figure A-3 (e,f)) exhibit clear and well-separated clusters for both datasets, indicating strong visual feature representation. In contrast, poor clustering performance is observed when only the text-based embeddings generated by the BLIP model are used (Figure A-3 (g,h)). This suggests that while BLIP provides valuable semantic information, its textual embeddings alone may not be sufficiently discriminative for the task of action recognition without the support of visual cues. As illustrated in Figure A-3 (a,b), the *FLAMeST* embeddings for UCF101 and HMDB51 form *well-separated and compact clusters, indicating that the combined representation effectively distinguishes between the two selected classes.* This demonstrates the discriminative power of the generated embeddings.

### C.2 Semantic Interpretability of VLM-Generated Descriptions.

Figure A-4 presents sample captions generated by the BLIP model. For each action class, a representative caption was produced using a language model for the CLIP model by prompting: "generate CLIP-suitable caption for the action". Whereas for the BLIP model used in *FLAMeST* the prompt was Describe the action in the image. In Figures A-4 (a) and (b), the captions align well with the visual content and accurately reflect the ground truth actions, indicating effective semantic understanding. However, Figure A-4 (c) illustrates examples where the generated captions are noisy, incomplete, or ambiguous, highlighting the limitations and failure cases of the captioning model in certain scenarios.

## D Failure Case Analysis

In this section, we analyze the representative failure cases of our method on the UCF101 and HMDB51 datasets. We also propose solutions that can mitigate the issues.

### D.1 Identification of Reasons for the Failure

We observe that broadly, these errors fall into three main categories: (A) selection of uninformative frames, (B) high inter-class similarity, and (C) noisy or incomplete caption VLM. Figure A-5 and Figure A-6 illustrate misclassified examples that exemplify the above failure modes.

- **Uninformative Frame Selection:** In *FLAMeST*, frames are sampled randomly from the video clips. Consequently, the selected frame may not adequately capture the action being performed. For instance, as shown in Figure A-5-A, the frames corresponding to the "kayaking" and "haircut" classes fail to depict critical visual cues such as a kayak or scissors—objects that are central to recognizing the activity. Similarly, for "wave" and

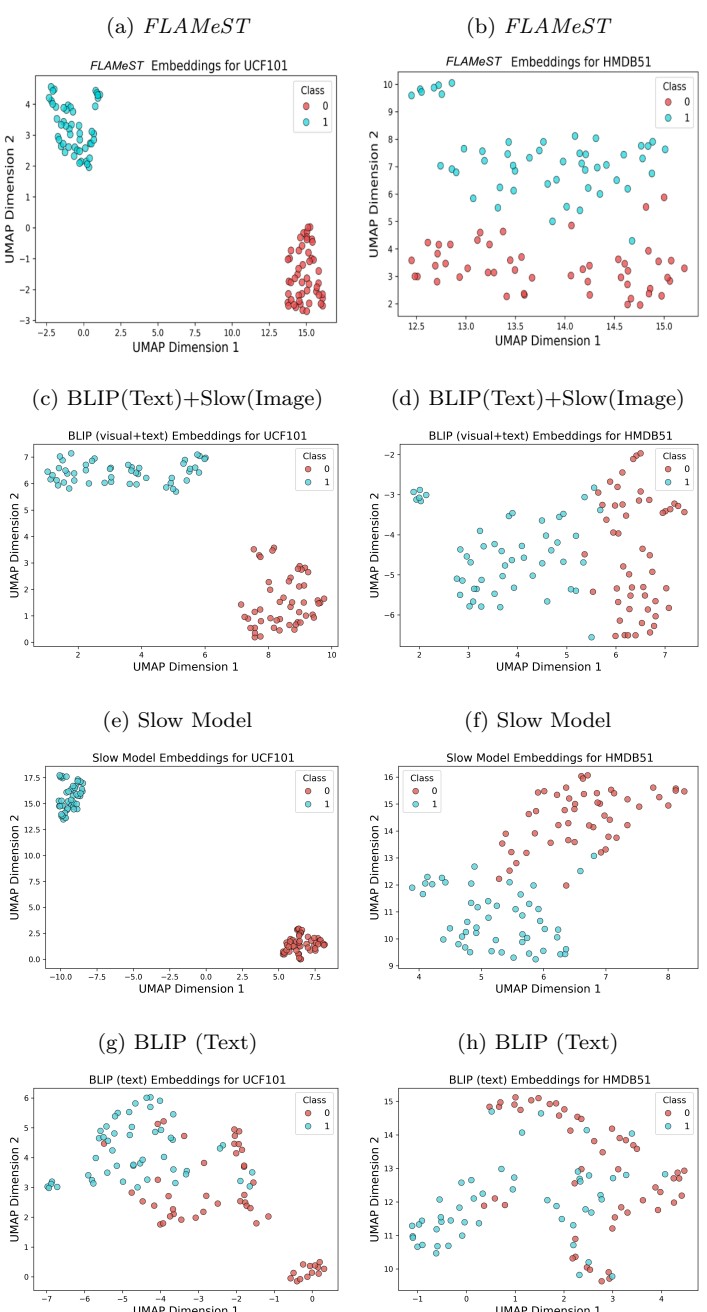

Figure A-3: UMAP visualization of different embedding strategies on selected classes from the UCF101 and HMDB51 datasets. For UCF101, we consider the classes 'Apply Lipstick' and 'Archery', For HMDB51, the selected classes are 'Catch' and 'Cartwheel.'

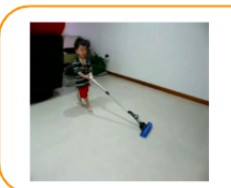 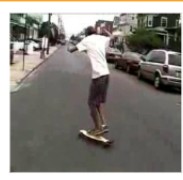 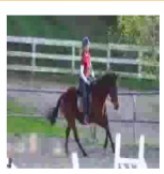

| Ground Truth | Moping | Skateboarding | Horse Riding |
|---|---|---|---|
| **BLIP MODEL** | A man is cleaning the floor with a mop. | A man riding a skateboard down a street. | A person riding a horse on a track. |
| **CLIP MODEL Text prompts generated by ChatGPT** | A person cleaning. | A person skateboarding on a ramp. | A person riding a horse. |

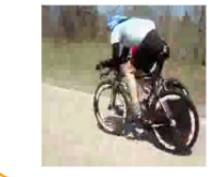 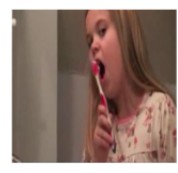 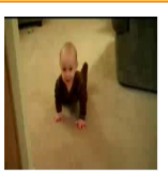

| Ground Truth | Bicycling | Brushing | Baby Crawling |
|---|---|---|---|
| **BLIP MODEL** | A person riding a bike down a street. | A girl is brushing her teeth in a mirror. | A baby is sitting on the floor in a room. |
| **CLIP MODEL Text prompts generated by ChatGPT** | A person riding a bicycle . | A person brushing their teeth. | A baby crawling on the floor. |

Figure A-4: Each example demonstrates the BLIP model's ability to generate natural language descriptions based on visual input. These auto-generated captions reflect the model's understanding of visual semantics and offer insight into its alignment between image and text domains. Class-specific captions precomputed by a large language model (LLM) are also presented for CLIP. These LLM-generated captions were curated specifically for compatibility with the CLIP model and serve as standardized textual prompts representing each action class.

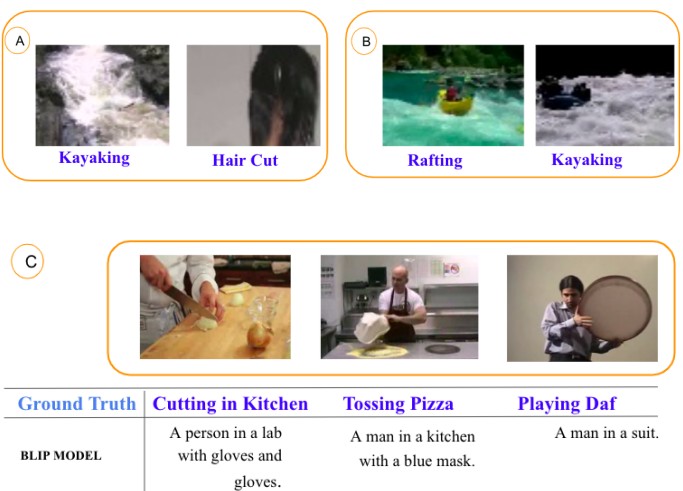

Figure A-5: The above figure represents failure cases from the UCF101 dataset. (A) Incorrect predictions due to the selection of uninformative frames that lack discriminative visual cues. (B) Confusion between visually similar action classes. (C) Noisy or incomplete captions generated by the VLM lead to misclassification.

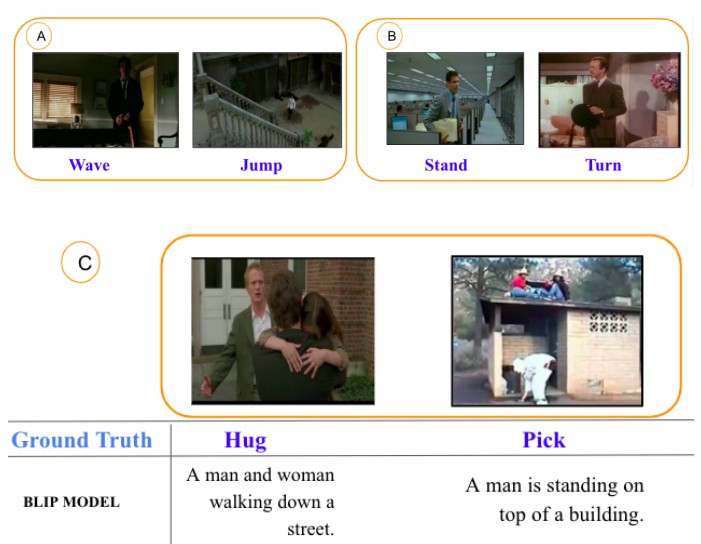

Figure A-6: The above figure illustrates failure cases from the UCF101 dataset. (A) Misclassifications arise from the selection of frames lacking salient visual information. (B) Ambiguities are caused by high visual similarity between action classes. (C) Errors are due to incomplete or inaccurate captions produced by the vision-language model.

"jump" in HMDB51 in Figure A-6-A, the selected frames show a static posture with no visible hand or jumping motion, leading to ambiguity in semantic interpretation.

- **High Inter-Class Similarity:** Certain action classes exhibit substantial visual overlap, particularly when temporal context is omitted. For example, "kayaking" and "rafting" both involve similar water-based environments and the presence of boats, making static frame-based distinction challenging (Figure A-5-B). Likewise, actions like "stand" and "turn" in HMDB51 (Figure A-6-B) can appear visually similar in still frames, especially if motion or orientation changes are not captured.

- **Noisy or Incomplete Captions:** Another source of error occurs from the VLM's inability to generate accurate and descriptive captions for the selected frames. As depicted in Figure A-5-C and A-6-C, the generated captions often fail to align with the image's semantic content or lack essential information.

### D.2 Proposed Solutions to Address Failure Cases

To mitigate the challenges identified in the failure analysis, we propose the following enhancements to the current framework:

- **Improved Frame Selection Strategy:** In the current pipeline, a randomly selected keyframe from each video is used as input to the VLM. However, this can result in uninformative or contextually poor frames that lack salient action cues (Figures A-5-A and A-6-A). To address this, a more effective frame selection mechanism is essential. One potential solution is to sample multiple informative keyframes and pass each individually through the VLM, subsequently averaging the resulting embeddings. This approach reduces the likelihood of omitting critical visual elements and enhances the robustness of representation.

- **Incorporating Temporal Information:** Given that many action classes (e.g., "walk," "stand," "turn") are temporally dynamic in nature, relying solely on static frame inputs limits the model's ability to distinguish between actions with similar spatial characteristics (Figure A-6-B). This limitation is particularly evident in the HMDB51 dataset, where *FLAMeST* demonstrates relatively lower performance. Integrating temporal features—extending the VLM to handle short clips or fusing VLM outputs with motion-aware embeddings—could significantly enhance discriminative power.

- **Fine-tuning the VLM:** Many of the captioning failures are attributed to the generic pre-training of the VLM on web-scale image-text datasets, which may not cover domain-specific cues found in action datasets. Fine-tuning the VLM on a curated set of video frames with high-quality action descriptions can help generate more accurate and semantically aligned captions.

## E  Extension beyond Action Recognition

We have extended the idea of leveraging vision–language models (VLMs) for image classification within a federated learning framework. This represents another line of our ongoing work. For each

image, a caption is first generated using the BLIP model. The image and its corresponding caption are then jointly processed by BLIP to obtain cross-modal embeddings. Each client employs a frozen VLM to extract these embeddings, which are subsequently used to train local MLP models. The parameters of the MLPs are transmitted to the server for weighted aggregation, and the resulting global model is shared back with the clients to complete the federated learning cycle. We tested our method on CIFAR-10 across different configurations (Figure A-7).

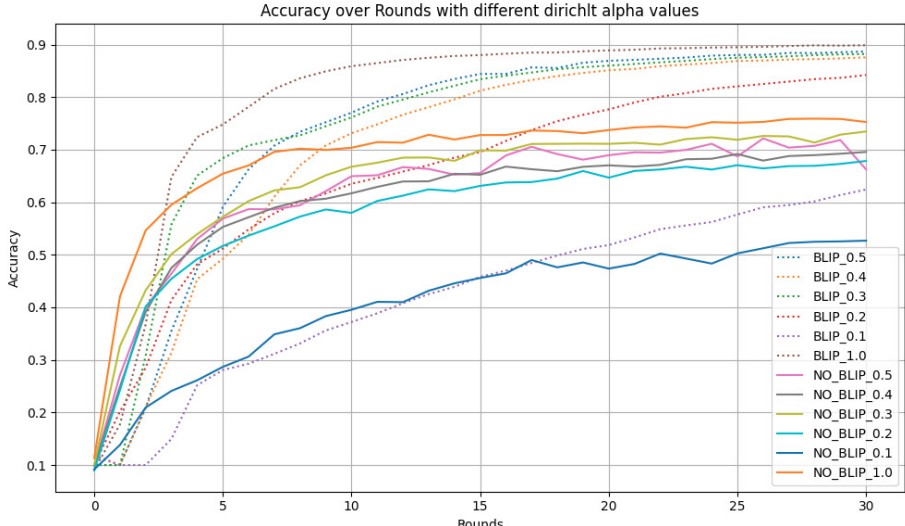

Figure A-7: The figure shows the accuracy of a global model achieved in an FL set-up using different configurations of the BLIP model. The x-axis shows the number of rounds, and the y-axis denotes accuracy.