# OpenReview forum: "Federated Multimodal Fusion for Action Recognition Leveraging Vision-Language Embeddings and Spatio- Temporal CNNs"
_TMLR — Accepted by TMLR_

### Review · Reviewer_VnGY · 2025-08-19

**Summary Of Contributions:**

This work proposes a Federated Learning (FL) framework for Action Recognition (AR) models, leveraging cross-modality embeddings (semantic, image, and temporal) and partial parameter aggregation to enhance efficiency. The experiments validate the effectiveness of the proposed embeddings, the computation efficiency, and effectiveness of the FL.

**Audience:**

Yes

**Audience Explanation:**

The paper represents a solid contribution to efficient FL for AR, with rigorous experimentation.

However, concerns persist about its practical relevance:
- The work risks positioning FL as an end in itself. Given the rise of high-performing open-source foundation models, the paper does not sufficiently justify why FL is a superior or complementary direction for privacy-sensitive video understanding.
- Leveraging private data via FL may not outperform strategies that enhance zero-shot performance using public data, especially given the flexibility of prompting/in-context learning in modern models. The absence of evidence positioning FL as a necessary tool for cutting-edge video understanding weakens its perceived significance.

**Broader Impact Concerns:**

No Specific Concerns

**Claims And Evidence:**

Yes

**Claims Explanation:**

- The paper is well written, with clear exposition of methodology and objectives.
- Empirical validation is thorough, featuring abundant experiments and ablation studies that robustly support the authors’ claims.
- The technical approach is novel in combining cross-modality embeddings with partial FL parameter aggregation, addressing a gap in efficient FL for video understanding.

**Requested Changes:**

- I am curious on how does FL-based AR fare against scalable open-source models (e.g., Internvideo2 [1]) that achieve strong performance without privacy constraints. The paper mentioned ACTION-CLIP only, which is a little bit out-of-date.
- A key FL advantage lies in distributed data scaling. However, experiments fix the overall dataset size while varying client numbers. To better assess scalability, analyzing performance as client count increases (with per-client data fixed) would clarify whether FL’s benefits scale under realistic distributed settings.
- The framework is framed as a general approach for video understanding, validation is limited to AR. Expanding discussions—or experiments—on its applicability to broader tasks (e.g., event recognition, video captioning) would strengthen claims of generality.

---

### Review · Reviewer_ioup · 2025-09-11

**Summary Of Contributions:**

In this paper, the authors propose a new method for federated learning for video-based action recognition (VAR) that leverages modern visual-language models (e.g., BLIP) to generate more detailed and discriminative embeddings. At its core, the method combines several ideas: 1) extracting embeddings from given video clips with pre-trained foundational models can dramatically decrease the number of weights that need to be exchanged during the federated learning step, and 2) in contrast to previous VAR approaches that leveraged VLMs, using only captions for training an action classifier is not optimal; therefore, combining both text-based caption embeddings and visual embeddings yields better performance. The authors benchmark their method on a number of popular action recognition datasets and demonstrate the effectiveness and efficiency of the proposed approach.

Weaknesses:

* The contribution of the method is mostly interesting from an engineering perspective, as it primarily demonstrates that modern pre-trained foundational and vision-language models can already produce semantically rich and useful embeddings that can be successfully utilized in downstream tasks such as video action recognition. While this is a valuable observation from a practical standpoint, the paper does not provide substantial novel theoretical insights or methodological innovations beyond adapting existing models, which may limit its impact from a research perspective. Nevertheless, acknowledging this engineering angle, the paper remains valuable in highlighting the applicability and efficiency of such embeddings in federated learning setups.

* The method contains a step where one selects key frame(s) from the video and generates captions for them (followed by generating embeddings with a VLM cross-encoder). This particular step feels somewhat ad hoc. One potential alternative would be to use video-language models (such as Qwen-VL or Cosmos-Reasoning variants) that generate textual descriptions for the entire video and also support embedding generation with cross-attention. It would be very interesting to see how such an approach performs, since the (semi)manual selection of key frames might severely underperform, while a caption for the entire video should describe the action much more precisely.

Strengths:

* The point about using text captions for describing complex video actions not being sufficient is interesting and useful for a broader audience. In a nutshell, with this claim the authors suggest that simple text-based classification/retrieval for videos is underperforming and that additional information from the visual component is necessary.

* The idea of using pre-trained models that can be shared across different clients is both practical and useful, as it significantly reduces the total amount of weights that need to be transferred during training. This not only lowers communication costs in federated learning setups but also makes the approach more scalable and applicable in real-world scenarios where bandwidth and efficiency are critical.

**Audience:**

Yes

**Audience Explanation:**

The paper will be of interest to members of the TMLR audience working on federated learning, video-based action recognition, and the integration of large pre-trained models into downstream tasks. The findings highlight practical ways to reduce communication costs in federated learning by leveraging pre-trained embeddings and show that relying solely on text captions for video understanding is insufficient (which popular approaches to the task do).

**Claims And Evidence:**

Yes

**Claims Explanation:**

The claims are supported by empirical evaluation on several standard video action recognition datasets, which demonstrates both the effectiveness and efficiency of the proposed approach. The authors clearly show that combining visual embeddings with caption-based embeddings outperforms using captions alone, and they provide benchmarks to substantiate this claim. The results also convincingly illustrate the practicality of reducing communication costs in federated learning through the use of shared pre-trained models.

**Requested Changes:**

* The current choice of selecting key frames manually (or semi-manually) for caption generation feels ad hoc. The authors should provide a clearer rationale for this choice or evaluate an alternative—e.g., using video-language models (such as Qwen-VL or Cosmos-Reasoning variants) that generate captions for entire videos—to ensure this step is not a weak link in the method.

* While modern VLMs generally produce good captions, for very complex behaviors their descriptions may lack sufficient detail to accurately represent the action, which can lead to weaker text embeddings (especially when using only one frame for captioning). This limitation should be explicitly discussed.

* In addition to the previous comment, the current evaluation does not clearly show how accurate the generated captions for the key frames are. Do they truly describe the observed action in the video? Demonstrating this would make it clearer that the method works as intended.

* One of the recent papers [1] suggests that using only text-based retrieval is suboptimal and that additional strategies—such as multi-vector retrieval—are needed. These findings are aligned with the observations in the current paper (e.g. using both visual + text-based embeddings) and should be explicitly discussed to place the work in the context of ongoing research.

[1] Weller, Orion, et al. "On the Theoretical Limitations of Embedding-Based Retrieval." arXiv preprint arXiv:2508.21038 (2025).

---

### Review · Reviewer_zJ5z · 2025-09-24

**Summary Of Contributions:**

This paper proposes FLAMeST, a federated learning framework for video action recognition that integrates BLIP and spatiotemporal CNNs to balance privacy, efficiency, and performance. Unlike prior works that use BLIP only for captioning, FLAMeST fuses semantic and motion embeddings to train a lightweight MLP, sharing only MLP parameters during aggregation with FedAvg, thereby preserving privacy and reducing communication overhead by 99%. Experiments on UCF101 and HMDB51 show significant performance gains over baselines (+5.13% and +2.71% accuracy).

**Additional Comments:**

No.

**Audience:**

Yes

**Audience Explanation:**

Yes. People working in the Video Action Recognition domain may be interested in this paper.

**Broader Impact Concerns:**

N/A.

**Claims And Evidence:**

Yes

**Claims Explanation:**

Most of claims are accurate

**Requested Changes:**

1. Figure 1: The FL procedure should be explicitly illustrated. Since Figure 1 presents the FLAMeST pipeline, it should clearly show how federated learning integrates with the video action recognition (VAR) component.

2. Table (1): The comparison methods are outdated. The authors should consider including more recent approaches from 2024 and 2025 to provide a stronger baseline.

3. Tables (2) and (3): These appear to be ablation studies. However, the direct comparison between the proposed FLAMeST framework and the baselines listed in Table (1) is missing and should be added.

---

### Author Response · Authors · 2025-12-07
**Author Global Response**

We thank the Action Editor and reviewers for their detailed and constructive feedback throughout the review process. In this final revision, we have:

1) Completed and clarified the Introduction, including a clearer motivation and positioning of FLAMeST within parameter-efficient FL with foundation models.

2) Corrected citation formatting and fixed layout issues in tables that previously exceeded page margins.

3) Removed unnecessary underlining in the references by adjusting hyperlink settings.

4) Added a new section, “Broader Applicability,” which explicitly discusses how FLAMeST can generalize to additional tasks, datasets, and backbone configurations beyond the video action recognition setting studied in our experiments.

5) The revised main manuscript and supplementary material have been uploaded.

6) Point-by-point responses to earlier reviewer comments remain available in the individual reviewer threads and are consistent with this final revision.

---

> ### Author Response · Authors · 2025-12-27
> **Author Global Comment**
>
> We have uploaded the final, deanonymized camera-ready version of the manuscript and supplementary material, incorporating all minor revisions requested by the Action Editor. Reviewer-specific change highlights have been removed.

---

### Decision · Action_Editor_mqHx · 2025-11-27

**Recommendation:** Accept with minor revision

**Additional Comments:**

After the authors' revision, most of the reviewers' concerns have been addressed. The reviewers unanimously recommend acceptance of the paper.

I find the work valuable and recommend its acceptance and publication.

I have the following suggestions for the authors as they prepare the final version:

* Although the authors have restored the missing Introduction section, it still appears incomplete. For instance, the third and fourth paragraphs seem unfinished.


* The citation format needs attention. In several places, there is misuse of \\citep and \\citet, and spaces are missing between citations and the preceding text.


* Some tables require correction. For example, Tables 6 and 8 extend beyond the main text margins.


* Underlining in the reference section is unnecessary and should be removed.


* Regarding the remaining concern from Reviewer VnGY – whether focusing on federated learning (FL) for a single layer, specifically within the task of activity recognition, aligns with the broader trends and technical developments in the field. Exploring FL across a more diverse set of tasks, open datasets, and model configurations could further strengthen the contribution – it would strengthen the paper to include a discussion addressing how the approach might generalize to additional tasks, datasets, and model configurations.

**Audience:**

Yes

**Audience Explanation:**

This paper may be of interest to researchers and practitioners working on federated learning, video action recognition, video understanding, and the use of pre-trained foundation models for downstream tasks.

**Claims And Evidence:**

Yes

**Claims Explanation:**

Claims And Evidence:

Summary:

This paper introduces a federated learning framework for video action recognition that systematically investigates how multimodal embeddings and partial model aggregation can improve privacy, communication efficiency, and performance in distributed settings. The method leverages cross-modality features by combining semantic embeddings from a vision-language model (BLIP) with spatiotemporal motion embeddings extracted from a pre-trained 3D CNN. These complementary representations are fused into a unified feature space to train a lightweight MLP, enabling federated learning where only the MLP parameters are shared, keeping raw videos and generated captions local. This design reduces communication overhead compared to full-model federated training while preserving privacy. Controlled evaluations on UCF101 and HMDB51 demonstrate that integrating both semantic and motion cues yields consistent gains over prior approaches that rely solely on caption-based features. Overall, the study provides evidence that multimodal embeddings and selective parameter sharing offer an effective and efficient pathway for scalable federated video action recognition.


Claims:

The paper makes several key claims: (1) combining features from pre-trained foundation models yields semantically rich and discriminative embeddings for video action recognition, leading to improved accuracy and efficiency in federated settings; (2) selective parameter sharing within the federated learning cycle substantially reduces communication overhead compared to transmitting the full foundation model, while also preserving data privacy; and (3) the proposed approach is robust to different client-update schemes and remains scalable across diverse and potentially non-IID data distributions.


Evidence:

The empirical evidence presented in the paper largely supports the claims, and the study’s limitations are acknowledged and addressed in the revision.

---

> ### Author Response · Authors · 2025-12-07
> **Response to Decision by Action Editor mqHx**
>
> We thank the Action Editor for the careful reading, positive assessment, and concrete suggestions for the final version. We have addressed all the requested minor revisions as follows.
>
> 1) Introduction completion: The third and fourth paragraphs of the Introduction have been revised to provide a complete motivation and a clear lead-in to our contributions, including an explicit positioning of FLAMeST within parameter-efficient FL with foundation models.
>
> 2) Citation formatting: We have corrected citation usage throughout the manuscript (consistent use of \citep) and fixed missing spaces between citations and the preceding text.
>
> 3) Table layout: Tables that previously exceeded the page margins (notably Tables 6 and 8) have been reformatted to fit within the text width.
>
> 4) Reference styling: Underlining in the reference section has been removed by adjusting the hyperlink settings, so that references now follow the journal style.
>
> 5) Discussion of generalization: Following your suggestion, we added a dedicated section titled “Broader Applicability,” where we discuss how the proposed FLAMeST framework can generalize beyond video action recognition to other tasks (e.g., event detection, retrieval, gesture recognition), datasets, and backbone configurations (including stronger and heterogeneous encoders), and how this aligns with broader trends in parameter-efficient federated learning with foundation models.
>
> We appreciate the constructive feedback and believe these changes improve both clarity and positioning of the work.